# LARGE LANGUAGE MODELS HAVE INTRINSIC SELF-CORRECTION ABILITY

## ABSTRACT

Large language models (LLMs) have attracted significant attention for their exceptional abilities in various natural language processing tasks, but they suffer from hallucinations that will cause performance degradation. One promising solution to improve the LLMs' performance is to ask LLMs to revise their answer after generation, a technique known as self-correction. Among the two types of self-correction, intrinsic self-correction is considered a promising direction because it does not utilize external knowledge. However, recent works doubt the validity of LLM's ability to conduct intrinsic self-correction. In this paper, we present a novel perspective on the intrinsic self-correction capabilities of LLMs through theoretical analyses and empirical experiments. In addition, we identify two critical factors for successful self-correction: zero temperature and fair prompts. Leveraging these factors, we demonstrate that intrinsic self-correction ability is exhibited across multiple existing LLMs. Our findings offer insights into the fundamental theories underlying the self-correction behavior of LLMs and remark on the importance of unbiased prompts and zero temperature settings in harnessing their full potential.

## 1 INTRODUCTION

Large Language Models (LLMs) have transformed the field of natural language processing, showcasing exceptional capabilities in generating coherent, contextually appropriate text across diverse domains (OpenAI, 2024; Gao et al., 2024). As of now, while LLMs are still not considered equivalent to human-level intelligence (Shen et al., 2023; Mathur et al., 2024), they exhibit some interesting characteristics that are possessed by human beings, including the emergent ability (Wei et al., 2022a)[1] and performing complex tasks (Chiang & Lee, 2023; Gruver et al., 2023). One particularly compelling aspect of human cognition is the ability to self-correct. Self-correction (SC) is the continuous process of evaluating and refining one's responses or actions to achieve improved outcomes. There are generally two types of SC: *extrinsic* SC which utilizes the help of external knowledge and *intrinsic* SC which uses only one's own knowledge. Inspired by humans, the research community proposed a promising hypothesis that we could leverage the power of SC to enhance the quality of LLMs. In particular, researchers are interested in whether *intrinsic* SC ability could be applied in LLMs.

Over the last few years, numerous research studies have been conducted on the effectiveness of intrinsic SC of LLMs. For example, works like self-refine (Madaan et al., 2023), self-verification (Gero et al., 2023; Weng et al., 2023; Manakul et al., 2023), post-editing (Raunak et al., 2023), all demonstrate significantly enhanced performance of SC on respective benchmarks compared to results obtained without SC. However, recent works have questioned the efficacy of intrinsic SC mechanisms in LLMs (Huang et al., 2024; Kumar et al., 2024). A notable critique presented by (Huang et al., 2024) is that LLMs do not possess the ability to perform intrinsic self-correction. Their argument is based on a fundamental question: *why couldn't an LLM get the correct answer in the initial attempt instead of going through intrinsic SC?* They hypothesize that existing works on SC abilities in LLMs shall be attributed to three factors: external feedback (such as the existence of a retrieval model) that converts the problem to an extrinsic setting, ground-truth label as an early stopping criterion that hinders the practical usage, and uninformative prompts that limit LLM's ability when it generates the first response before SC (which raises the possibility of unfair comparison). Through the experiments,

---

[1]whether LLMs exhibit emergent ability is still in debate in academia, partially due to the definition of "emergence". See Schaeffer et al. (2023) and Lu et al. (2023) for details.

Figure 1: An example where the biased prompt (left) shifts the answer from correct to incorrect between the intrinsic SC stages, whereas our unbiased prompt (right) maintains the correct answer. The blue, yellow, and red regions correspond to Stage 1,2,3 in Section 2.2, respectively. Phrases such as "find problems" and "improve" might hint at an incorrect initial answer to the LLM and force it to change answers. On the other hand, the unbiased prompt avoids those unnecessary changes.

both Huang et al. (2024) and Kumar et al. (2024) report that LLMs showcase different levels of degradation in QA accuracy after intrinsic SC.

In light of such disagreement, it is important to answer a set of fundamental questions: **Does intrinsic SC ability exist within LLMs? And if it does, why couldn't an LLM get the correct answer in the initial attempt instead of going through intrinsic SC?**

This paper answers the above fundamental questions. With theoretical analyses and quantitative experiments under the same settings as Huang et al. (2024), we empirically show that LLMs **do** have intrinsic SC ability. In addition, we also show that LLMs cannot answer questions correctly during the initial attempt because of their inherent hallucination property.

Building on top of the answers, further investigations are conducted on the conditions that maximize SC potential within LLMs. Through theoretical analyses and empirical experiments, we identify two factors impacting the intrinsic SC ability of LLMs. The first factor is temperature. A non-zero temperature, such as done by Huang et al. (2024), degrades SC ability. The second factor, equally important, is the neutrality and fairness of the prompts used during various stages of SC. A fair and unbiased prompt that doesn't directly or indirectly influence the LLM to change or maintain its initial answer provides the most stable benefits from the intrinsic SC ability of LLMs. In Fig. (3), we provide an example illustrating how biased (Huang et al., 2024) and unbiased (ours) prompts affect the output of the LLM during different stages of SC.

This paper's contributions could be summarized as the following:

- We provide a novel perspective on the functioning nature of intrinsic self-correction in Section 2. We posit that self-correction abilities bear similarities to chain-of-thought (CoT) and self-verification prompting methodologies, which furnish LLMs with additional context for generating the final response.

- We highlight the importance of the temperature of LLMs in the success of intrinsic self-correction in Section 4. While increasing the temperature will benefit the creativity of response, it may be detrimental for SC, depending on the structure of LLM's response.

- Additionally, in Section 5, we show that the SC prompt needs to be unbiased so that LLMs do not accidentally attend to tokens that might trigger unwanted answer updates during the SC process. We also provide guidelines to compose unbiased prompts for future research.

- We empirically verify the effects of the two factors above, showcasing the accuracy degradation of generated SC responses when increasing temperature and utilizing biased prompts, and intrinsic SC can be achieved by LLMs via fair prompts and zero temperature.

Our answers to those fundamental questions serve as a basis for the underlying mechanism of intrinsic SC, and we hope that our findings will shed further light on LLM's abilities and encourage future works and discussions to explore methods that could leverage the existing SC ability more effectively.

## 2 NEW PERSPECTIVE ON INTRINSIC SELF CORRECTION

### 2.1 PRELIMINARY

The text generation process of LLMs could be modeled as an auto-regressive probabilistic distribution $P$. Given an LLM and a prompt of $t$ tokens $\tau = \{w_1, w_2, ...w_t\}$, the auto-regressive generation of the response takes the form of Eq. (1), where $y$ is the (unnormalized) logits from the LLM, $w_{t+1}$ is a potential next token, $l_{w_{t+1}}$ is the scaled logit value for $w_{t+1}$, and $T$ is the temperature. Thus, the generated content of an LLM adheres to the distribution of its logits, readjusted by the temperature. As temperature approaches 0, it is clear that the generation process changes from sampling in a distribution to selecting the index of the maximum value of logits, as described in Eq. (2).

$$P(w_{t+1}|w_1, ...w_t) \sim y \quad , \quad l_{w_{t+1}} = \frac{e^{\frac{y_{w_{t+1}}}{T}}}{\sum_{k=1}^{N} e^{\frac{y_k}{T}}} \tag{1}$$

$$\lim_{T \to 0^+} l_{w_{t+1}} = \begin{cases} 1 & \text{if } y_{w_{t+1}} = \max y \\ 0 & \text{otherwise} \end{cases} \tag{2}$$

Although the definitions above may appear elementary, they constitute critical foundations for this study. In particular, they provide justifications for some of the analyses in the future sections. We draw several insights on this foundational formation:

(1) Under $T = 0$, any arbitrary LLM should output the same response on some fixed prompt $\tau$. Unfortunately, models of the GPT family (OpenAI, 2024; Brown et al., 2020) do not follow this principle, possibly due to their query optimizations and internal sparse mixture-of-experts (MoE) design of GPT-4. In particular, GPT-4 exhibits very high variance for its answers under $T = 0$. Those unknown behaviors cause some variations in the evaluation process.

(2) Intrinsic SC increases the prompt length $t$ constructively, which helps improve the generation quality of the final response. This is similar to self-verification (Weng et al., 2023; Gero et al., 2023) and chain-of-thought (CoT) (Wei et al., 2022b). We will elaborate on this in Section 2.2.

(3) Increasing temperature might not cause visible accuracy degradation in the evaluation process for response generation without SC, but it might be detrimental to the intrinsic SC due to the non-optimal sampling process. We will elaborate on this in Section 4.

### 2.2 DOES INTRINSIC SC EXIST IN LLMS?

In this section, we answer the first fundamental question of whether intrinsic SC exists in LLMs by providing an alternative explanation of the underlying mechanism of self-correction. For now, we focus on one model (GPT3.5-0613) and fix the temperature at zero.

We initiate our analysis with the chain-of-thought (CoT) prompting technique [2], which is proven to be useful in enhancing LLM's performance in accuracy (Wei et al., 2022b). Based on the auto-regressive nature of LLM generation, Chu et al. (2023) models the CoT process as a conditional probability shown in Eq. (3), where A is the answer to the prompt (and question) $\tau_1$, and $R_1$ is the rationale provided by LLM in the CoT process. The reasoning context introduced by CoT helps the LLM make a better final decision. Similarly, using the effective depth of the transformer circuit, Feng et al. (2023) argues that CoT increases the effective depth of the circuit by letting the generated outputs repeatedly loop back to the input [3].

---

[2] CoT prompt could be elicited with phrases such as "explain your reasoning" or "think step by step" in LLMs during the prompting.

[3] It shall be noted that Feng et al. (2023) actually proves that with longer generations, transformers (for **some** choice of weights; not necessarily the pre-trained weights) have more expressive capacity to model more complex problems.

$$p(A, R_1|\tau_1) = p(A|\tau_1, R_1)p(R_1|\tau_1) \tag{3}$$

Similarly, it is apparent that intrinsic SC also has the potential to achieve CoT-like behavior. Generally, we could model the generation process of a typical intrinsic SC according to Eq. (4). The first stage is a generic CoT prompting and answering schema. In the second stage, based on the first part's results, we use a prompt $\tau_2$ to let LLM provide rationales ($R_2$) and a decision ($D$) on the correctness of the previous answer. Then, in the last stage, we instruct LLM with $\tau_3$ to give a final answer $A'$ based on all previous information. We refer readers to Fig. (3) for an example of each stage.

$$p_{SC}(A', D, R_2, A, R_1|\tau_3, \tau_2, \tau_1) =$$
$$\underbrace{p(A'|\tau_3, D, R_2, \tau_2, A, R_1, \tau_1)}_{\text{stage 3}} \underbrace{p(D, R_2|\tau_2, A, R_1, \tau_1)}_{\text{stage 2}} \underbrace{p(A|R_1, \tau_1)p(R_1|\tau_1)}_{\text{stage 1}} \tag{4}$$

The second stage is the key factor of the SC process. Since it requires the LLM to provide a decision over the correctness of the previous answer as well as the rationales, there are two different orderings for the output depending on the position of the decision and rationale. We could model those two orderings as a conditional probability distribution, as shown in Eq.(5) where **Order 1** refers to having the decision first followed by rationale and **Order 2** being the other way around:

$$Order\ 1: \underbrace{p(R_2|D, \tau_2, A, R_1, \tau_1)}_{\text{rationale}} \underbrace{p(D|\tau_2, A, R_1, \tau_1)}_{\text{decision}}$$
$$Order\ 2: \underbrace{p(D|R_2, \tau_2, A, R_1, \tau_1)}_{\text{decision}} \underbrace{p(R_2|\tau_2, A, R_1, \tau_1)}_{\text{rationale}} \tag{5}$$

Depending on the different prompts designs, we discuss the different behaviors of intrinsic SC below:

**Case 1: if $\tau_1$ uses CoT and the second stage follows Order 1**  In this case, intrinsic SC is just an implicit form of self-verification (Gero et al., 2023; Weng et al., 2023). Based on the rationales given in stage 1, stage 2 will provide an answer on the consistency of the provided rationales by reasoning on them. As shown in Table 1, there is an improvement in accuracy after SC. We will show more experiments in Section 5 that this case provides some accuracy improvement to the final result.

**Case 2: if $\tau_1$ uses CoT and the second stage follows Order 2**  In this case, SC could be seen as a CoT answer over a prompt that contains CoT-alike context. By providing more rationales in the second stage, SC provides additional analyses over the provided rationales in the first stage and makes a better decision. If well implemented, this case could lead to better performance compared to the first one. However, such an approach might not be very feasible as we cannot fully control the decomposition orderings of LLMs' output through prompts. During our experiments, GPT-3.5 shows a strong preference for the decomposition Order 1, and the designed prompt is only able to successfully change some responses' order.

**Case 3: if $\tau_1$ does not use CoT and the second stage follows Order 1**  When $\tau_1$ does not use CoT, the decomposition of the first stage in Eq. (4) is simply $p(A|\tau_1)$. In this case, intrinsic SC likely has a negative impact, and it might cause an accuracy drop after correction. This is because the decision is first given without any rationale, and subsequent provided rationales are now dependent on a possibly hallucinated answer.

**Case 4: if $\tau_1$ does not use CoT and the second stage follows Order 2**  This serves the same purpose as CoT, and thus accuracy will improve. By requiring LLMs to first provide rationales, followed by a decision on the answer, and finally offering a more informed conclusion, SC enhances the overall accuracy. The improved accuracy is comparable (if not matching) to merely using CoT prompt with no self-correction as shown in Table 1.

Hence, we conclude that intrinsic SC ability exists in LLMs with an adequate prompting style and is potentially as effective as other prompt engineering techniques. We provide an example of the prompt and outputs of each case in Appendix E respectively.

Table 1: Accuracy of GPT3.5 on CommonSense QA dataset for cases presented in Section 2.2.

| CommonSense QA | Before SC | After SC |
|---|---|---|
| Case 1 (CoT, Order 1) | 75.92 | 76.00 |
| Case 2 (CoT, Order 2) | 75.92 | 76.49 |
| Case 3 (no CoT, Order 1) | 75.35 | 74.77 |
| Case 4 (no CoT, Order 2) | 75.35 | 75.92 |

## 2.3 WHY CAN'T LLM ANSWER QUESTIONS CORRECTLY IN THE INITIAL ATTEMPT?

When LLMs hallucinate, one of the scenarios they exhibit is that they will output answers incorrectly even when having the correct knowledge. While that is not the full schema of LLM hallucination, in this paper, we simply refer to this type of scenario as "hallucination". Assuming LLMs have a non-zero possibility of hallucination, we show that LLMs will not reach their full ability in the initial attempt with Proposition. 2.1.

**Proposition 2.1.** *LLMs are generally under-performing compared to their true ability because hallucination will cause the overall accuracy to decrease.*

This lemma serves as a justification for the second fundamental question and we provide the proof of Proposition. 2.1 in Appendix C. As drawn by the conclusion in the previous section, intrinsic SC can reduce hallucinations of the generated responses and achieve better quality similar to other prompt engineering techniques.

## 3 EXPERIMENT SETUP

To better support our argument in the previous section, we conduct a set of quantitative experiments that study the SC ability of existing models across various tasks. In this section, we outline the experimental setup. This setup, along with further theoretical analyses, is used throughout the paper to provide a deeper understanding of the optimal conditions under which SC ability thrives.

**Benchmarks.** We utilize six benchmarks to showcase the SC ability across a variety of tasks: CommonSense QA (Talmor et al., 2019), GSM8K (Cobbe et al., 2021), MMLU (Hendrycks et al., 2021), HotPotQA (Yang et al., 2018), Big Bench (Srivastava et al., 2022), and SVAMP (Patel et al., 2021). The details and usages of the benchmarks are discussed in Appendix B.

**Models.** To showcase our claims, we conduct our experiments on a variety of models. We use GPT-3.5-Turbo (gpt-3.5-turbo-0125) and GPT-4-Turbo (gpt-4-1106-preview) provided by OpenAI (OpenAI, 2024). We also evaluate the performance of self-correction on Llama-3.1-8B-Instruct (Dubey et al., 2024), and Mistral-7b-instruct-v0.2 (Jiang et al., 2023). These models are run locally on a server equipped with 2x NVIDIA A100 (80GB PCIe) GPUs.

## 4 INCREASING TEMPERATURE MIGHT DECREASE ACCURACY DURING SC

As mentioned in Section 2.1 and Eq. (1), the temperature hyperparameter controls the randomness of the generated context. It is widely acknowledged within the research community that increasing the temperature parameter (within a reasonable range) enhances the creativity of LLMs, resulting in the generation of more diverse content (Hinton et al., 2015). Works like Wang et al. (2020) also try to dynamically optimize the temperature at the token level for best generation quality. Furthermore, a very recent work from Renze & Guven (2024) has claimed to find no correlation between increasing temperature and losing accuracy on benchmarks for temperatures between 0.0 and 1.0. However, as we will show in this section, temperature choice is a non-trivial aspect of the success of some LLMs during intrinsic SC. While increasing the temperature indeed shows no effect on the original response, it might have a negative impact on the subsequent stages of the SC process.

Our experimental results indicate that GPT-3.5 is adversely affected by increasing temperature, while GPT4 exhibits greater robustness to temperature variations. We provide the theoretical analysis below, based on the decomposition of the second stage of SC.

## 4.1 THEORETICAL ANALYSIS

We begin by demonstrating that the randomness in decision-making diminishes as the temperature decreases. We denote a positive decision ("yes") as $D = 1$, and vice versa $D = 0$ for a negative decision ("no"). Therefore, the decision procedure according to the context, which may include the prompts, rationales, and previous answers, degrades to a 0/1 binary classification that follows the binomial distribution.

$$p(D = 1) = \frac{\alpha_1^{\frac{1}{T}}}{\sum_i \alpha_i^{\frac{1}{T}}} = \frac{\alpha^{\frac{1}{T}}}{\alpha^{\frac{1}{T}} + (1-\alpha)^{\frac{1}{T}}}, \tag{6}$$

where $\alpha \in (0, 1)$ is the original positive decision probability of the model. Therefore, considering the variance of the binomial distribution, we have:

$$Var(D) = \alpha^{\frac{1}{T}}(1-\alpha)^{\frac{1}{T}}/(\alpha^{\frac{1}{T}} + (1-\alpha)^{\frac{1}{T}})^2 \tag{7}$$

Next, we consider the effect of $T$ going from 0 to $+\infty$:

$$\frac{\partial Var(D)}{\partial T} = \frac{\alpha^{\frac{1}{T}}(1-\alpha)^{\frac{1}{T}}[\ln(\alpha) - \ln(1-\alpha)][\alpha^{\frac{1}{T}} - (1-\alpha)^{\frac{1}{T}}]}{T^2 \left(\alpha^{\frac{1}{T}} + (1-\alpha)^{\frac{1}{T}}\right)^3} \geq 0, \tag{8}$$

The inequality holds because $(\ln(\alpha) - \ln(1-\alpha))$ and $(\alpha^{1/T} - (1-\alpha)^{1/T})$ share the same positive and negative sign, since we have $\alpha \in (0, 1)$ and $T > 0$. This implies that when there is an output preference ($0 < \alpha < 1$), the variance of the decision increases monotonically with the model's temperature, and the model with 0 temperature will be less likely to give the wrong decision due to less randomness. When the temperature goes to infinity, the model randomly selects an answer from {"yes", "no"} to give the decision, causing an unexpected decision error. At the same time, confidence in the decision is correlated to the variance: $\alpha = 0.5$ gives the highest variance of 0.25, whereas $\alpha$ approaching 0 or 1 will reduce variance.

With an increasing variance, $D$ has an increasing possibility of being flipped to the other side. Therefore, when $T$ is non-zero, a higher number of decisions will fail to reflect the true knowledge encoded within the LLM. In other words, those temperature-flipped binary decisions are in fact "hallucinations", and under Proposition 2.1, those hallucinations will cause accuracy decrements after the SC process.

Finally, we refer to Eq. (5) and examine the two decompositions. Through the analysis of the effect of $\alpha$ on the variance, we argue that Order 1 is more prone to temperature changes than Order 2. With the second ordering, the final decision after rationales is very close to a copying task, and LLMs are known to have very high confidence in performing this task ($\alpha$ very close to 0 or 1). On the other hand, providing the decision is a reasoning task that is much harder with less confidence ($\alpha$ closer to 0.5). Using Fig. (3) as an example, imagine a hypothetical answer such that GPT-3.5 puts the first sentence (decision) in stage 2 at the end. If so, to obtain the same information, it only needs to perform selective copying based on the rationales above, which is much easier (and thus more confident) than the current way.

## 4.2 IMPACT OF TEMPERATURE ON SELF-CORRECTION

We conducted experiments to investigate the influence of temperature on intrinsic SC performance using the CommonSense QA dataset (Talmor et al., 2019), using the optimized prompt set that will be elaborated in Section 5. The prompt remains fixed, with temperature adjustments applied across all three stages. As shown in Fig. 2a, increasing temperature causes the SC ability of GPT-3.5 to drop. GPT4 remains robust to temperature changes, and this is due to the structure of stage 2's answer to those models. Based on our observations, it seems that GPT-3.5 follows Order 1 of Eq. (5), whereas GPT4 follows Order 2. Since the second ordering has a much more deterministic decision after rationale, the variance of the decision when temperature increases is also much less than in the first one. We provide the responses of each model in Appendix F.

In addition, we conduct two ablation studies. First, we dissect the SC process to isolate the effect of temperature specifically on the intrinsic SC component. The response generated by LLM in the first stage under $T = 0$ is fixed, and the temperature is altered only for the second and third stages. As

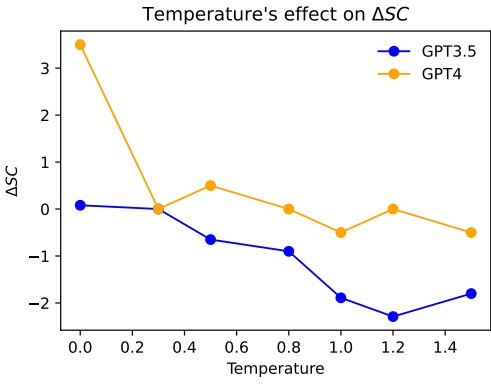 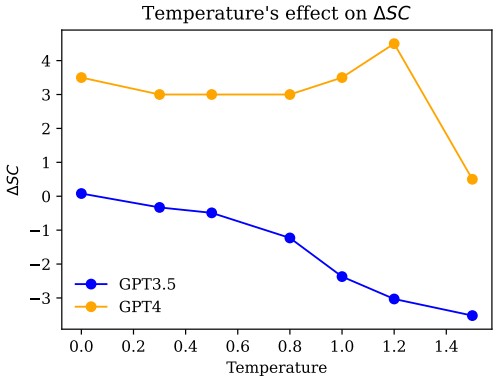

(a) There is an overall trend of LLMs losing SC ability as temperature increases, and GPT-3.5 is very vulnerable to such an increase in temperature.

(b) Results when the first stage is fixed. GPT-3.5's changes are clearer without variances in the first stage. GPT-4 remains robust.

Figure 2: Comparison of the effect of temperature on SC ability measured as the difference in accuracy before and after SC ($\Delta$ SC.) across the two GPT models on the Commonsense QA dataset.

shown in Fig. 2b, the same curve for accuracy decrement also appears on the CommonSense QA dataset with GPT-3.5. Without variations for the first part, GPT-3.5 shows an even clearer trend for accuracy decrement during the SC process when the temperature is increased. The detailed numerical results are provided in Appendix D.1. Second, we build upon the first ablation study and adopt the inadequate prompt used by Huang et al. (2024). While the results become much more inconsistent for the inadequate prompt, we see that as the temperature increases, there is still a general trend of LLM losing SC ability. We provide the detailed results for this ablation study in Appendix D.2.

## 5 DESIGNING FAIR PROMPT FOR SELF-CORRECTION

Prompt engineering plays a pivotal role in the generation process of LLMs. Numerous studies have demonstrated that carefully crafted prompts can substantially enhance the quality of the generated output (Wei et al., 2022b; Zhang et al., 2023; Weng et al., 2023). In this section, we demonstrate that prompt design significantly influences the SC performance of certain models and that a **fair** prompt is capable of eliciting *universal* intrinsic SC abilities in LLMs. In addition, we provide guidelines for fair prompting for future research in this direction.

To ensure a fair comparison, we focus exclusively on Case 1 as described in Section 2.2. Case 1 shares the same settings as Huang et al. (2024), with the only distinction being the semantic bias of the prompt. Empirical experiments are carried out by retaining the prompt from the first stage of intrinsic SC, adhering to the same prompt used by Huang et al. (2024), and demonstrating that inadequate prompts can lead to erroneous behavior in LLMs.

### 5.1 THEORETICAL ANALYSIS

Referring back to Eq. (4), we see the final output from stage 3 depends on eight factors. Among them, $A$, $R_1$, and $\tau_1$ are fixed from stage 1. Excluding those fixed constants, $D$ and $R_2$ depend solely on $\tau_2$. Therefore, we conclude that the final answer is contingent only upon two variables, $\tau_2$ and $\tau_3$, corresponding to the prompts in stages 2 and 3.

The analysis of the prompt's effect shares the same principle as Proposition 2.1. We investigate the effect of the biased prompts from Huang et al. (2024) and Kumar et al. (2024) that encourage the model to change the answer on the final result. We assume that the LLM strictly adheres to the prompt, randomly altering $\gamma\%$ of the answers that would remain unchanged under a fair prompt, where $\gamma > 0$[4].

---

[4]We assume $\gamma > 0$ for a reasonable LLM. From the semantic understanding, encouragement of changing answers should lead to more changes in the answer. We also provide the empirical number of changes of the answers in Appendix D.3.

The randomness of the changes is expected, as those are the cases in which LLMs would not have changed in a normal setting, and thus they should not be attributed to certain characteristics, such as correctness. For ease of analysis, we assume that LLM does not change any answer with a fair prompt (thus the accuracy difference after SC is 0).

Table 2: We evaluate several LLM models on six datasets across the three prompt sets that are discussed in Section 5.2. Intrinsic SC *universally* improves the accuracy of the four models with fair prompts (Set 3), whereas biased prompts (Set 1) might decrease the accuracy. On the averaged results, SC with the fair prompts shows improvement in response accuracy with statistical significance with respect to the number of changed responses ($p < 0.05$ for ** and $p < 0.003$ for ***).

| CommonSense QA | | | | |
|---|---|---|---|---|
| Model | Before SC | Prompt Set 1 | Prompt Set 2 | Prompt Set 3 |
| GPT3.5 | 75.92 | 66.34 | 72.56 | 76.00 |
| GPT4 | 84.11 | 73.55 | 82.80 | 84.60 |
| Mistral | 66.01 | 66.83 | 66.67 | 66.67 |
| Llama3.1 | 75.18 | 62.65 | 73.30 | 75.27 |
| HotpotQA | | | | |
| Model | Before SC | Prompt Set 1 | Prompt Set 2 | Prompt Set 3 |
| GPT3.5 | 31.00 | 22.00 | 45.00 | 47.00 |
| GPT4 | 88.00 | 80.00 | 86.00 | 89.00 |
| Mistral | 47.00 | 47.00 | 49.00 | 50.00 |
| Llama3.1 | 55.00 | 55.00 | 54.00 | 60.00 |
| MMLU (Formal Logic and Conceptual Physics) | | | | |
| Model | Before SC | Prompt Set 1 | Prompt Set 2 | Prompt Set 3 |
| GPT3.5 | 60.94 | 59.56 | 61.77 | 62.33 |
| GPT4 | 83.66 | 78.95 | 83.93 | 84.76 |
| Mistral | 44.04 | 48.48 | 45.15 | 44.88 |
| Llama3.1 | 66.76 | 61.25 | 64.54 | 67.31 |
| Big Bench (Sports) | | | | |
| Model | Before SC | Prompt Set 1 | Prompt Set 2 | Prompt Set 3 |
| GPT3.5 | 83.25 | 50.76 | 68.02 | 88.83 |
| GPT4 | 90.86 | 56.85 | 78.68 | 90.86 |
| Mistral | 61.93 | 61.93 | 61.42 | 62.94 |
| Llama3.1 | 78.17 | 52.79 | 76.65 | 78.68 |
| GSM8K | | | | |
| Model | Before SC | Prompt Set 1 | Prompt Set 2 | Prompt Set 3 |
| GPT3.5 | 77.63 | 78.24 | 78.17 | 77.94 |
| GPT4 | 94.5 | 89.5 | 92 | 95 |
| Mistral | 46.78 | 47.08 | 47.46 | 47.92 |
| SVAMP | | | | |
| Model | Before SC | Prompt Set 1 | Prompt Set 2 | Prompt Set 3 |
| GPT3.5 | 84.00 | 79.00 | 84.00 | 84.33 |
| GPT4 | 91.33 | 87.00 | 87.67 | 92.00 |
| Mistral | 59.33 | 59.00 | 59.00 | 59.67 |
| Llama3.1 | 80.00 | 69.33 | 77.00 | 84.00 |
| **Average across Datasets** | | | | |
| Model | Before SC | Prompt Set 1 | Prompt Set 2 | Prompt Set 3 |
| GPT3.5 | 74.84 | 69.07 | 73.50 | **75.93***** |
| GPT4 | 89.09 | 80.52 | 86.66 | **89.65**** |
| Mistral | 55.15 | 55.97 | 55.75 | **56.06***** |
| Llama3.1 | 73.79 | 62.10 | 71.78 | **74.76***** |

- **when $\tau_2$ is biased.** Since $D$ depends on $\tau_2$, $\gamma\%$ of $D$ will be inverted. Similar to the argument in Section 4, randomly inverting $D$ will lead to an accuracy drop. Also, $A'$ largely depends on $D$, we could say that roughly $\gamma\%$ of $A'$ will also be changed, with more correct

answers changing to incorrect versus the other side. The effect of randomly changing answers for $A'$ has the same proof as Proposition 2.1.

- **when $\tau_3$ is biased.** The extra $\gamma\%$ of changes could be seen as hallucinations that are forced upon LLMs. Then, the analysis follows the argument in Proposition 2.1. Since the answers before SC are better than random guessing, after randomly re-assigning for $\gamma\%$ of the responses during the SC process, more incorrect answers will appear after the assignment.

- **when both $\tau_2$ and $\tau_3$ are biased.** The detrimental effects of the two $\gamma\%$ changes accumulate.

## 5.2 Intrinsic SC could be achieved with fair prompt under zero temperature

We consider three sets of prompts. The first set (**Problem Set 1**) is used by Huang et al. (2024), but we argue that they are biased towards encouraging the LLMs to change answers due to the use of the negative phrase "find problems". The second set (**Problem Set 2**) is derived from the first one by removing the negative phrases and asking the LLM to check if the answer is correct or incorrect and act accordingly. While such a prompt may appear fair from a human perspective due to the balance of both aspects, the inclusion of the terms 'correct' and 'incorrect' can indirectly influence the LLM's behavior. Therefore in the last set of prompts (**Problem set 3**), we strip the prompts from such phrases and words to make a completely unbiased prompt. We provide details on the three sets in Appendix E. As shown in Table 2, using an unbiased prompt and zero temperature, all of the benchmarked LLMs exhibit accuracy improvement on the different benchmarks. Nevertheless, biased prompts may decrease accuracy in certain LLMs and specific benchmarks. It is noteworthy that this decrease seems to be related to some specific features of the benchmarks and models, but as of now, a reasonable explanation and prediction of the specific features are yet to be discovered. For example, in CommonSense QA benchmark (Talmor et al., 2019), the bias of the prompt has significant impacts on GPT and Llama models, but not so much on the other models. On the other hand, the bias of the prompt seems to only affect the GPT models, but it has no impact on the Mistral and Llama models for the HotpotQA benchmark (Yang et al., 2018). We hypothesize that this might be because those models are instruction fine-tuned, and GPT families are further fine-tuned with reinforcement learning with human feedback (RLHF) (Ouyang et al., 2022). Depending on the different training datasets that are not always disclosed, an LLM's accuracy after SC might be conditionally robust to biased prompts, depending on the context.

Importantly, our results align with the self-verification results obtained by Weng et al. (2023) where they also obtained a slight improvement using a different model. The on-par results further validate our claim in Section 2.2 that intrinsic SC is similar to other existing prompt engineering techniques.

In Appendix D.3, we provide further numerical analysis of how some of the models change their answers during SC. The results not only align with the theoretical analysis but also provide further insights into why some models remain robust on some benchmarks. With the $\gamma > 0\%$ extra changes caused by the biased prompt, more incorrect answers are introduced in the final stage.

All of the above analyses, reinforce our initial claim that a fair prompt is a safe option. A truly unbiased prompt must avoid both semantically and literally negative or positive phrases, such as "find the problem" or "correct your answer". **We encourage future researchers to follow these guidelines when composing their future SC prompts, similar to the unbiased Prompt Set 3.**

## 5.3 Ablation study on Model size

In addition to Table 2, we conduct an ablation study about the effect of model size under the effect of intrinsic self-correction using the Qwen 2.5 family (Team, 2024). As we can see from Table 3, the conclusion that intrinsic SC universally exists in LLMs remains true. Under the fair prompt (prompt set 3), all of the models exhibit a significant increase in accuracy on the benchmarks after SC. In addition, based on these preliminary results on the effect of model size, it seems that larger models have better intrinsic SC ability, as. This is intuitive, as larger models possess better abilities in understanding and following the prompts. However, we shall note that the results of this study are still very limited, and further investigations are needed to make a definitive claim.

Table 3: We evaluate three benchmarks related to math and logic on the Qwen 2.5 family to study the model size's effect on intrinsic self-correction.

| GSM8K | | | | |
|---|---|---|---|---|
| Model | Before SC | Prompt Set 1 | Prompt Set 2 | Prompt Set 3 |
| 3B | 85.22 | 85.22 | 85.32 | 85.52 |
| 7B | 83.55 | 87.19 | 88.55 | 88.93 |
| 14B | 74.53 | 86.2 | 86.13 | 88.55 |
| MMLU (Formal Logic and Conceptual Physics) | | | | |
| 3B | 57.34 | 57.06 | 58.17 | 59 |
| 7B | 69.25 | 64.27 | 67.31 | 70.36 |
| 14B | 66.94 | 71.19 | 68.98 | 71.75 |
| SVAMP | | | | |
| 3B | 88.33 | 87.00 | 88.00 | 88.67 |
| 7B | 87.67 | 86.33 | 90 | 91.33 |
| 14B | 78.00 | 82.00 | 88.33 | 87.67 |
| Average | | | | |
| 3B | 80.61 | 80.36 | 80.78 | 81.16 |
| 7B | 81.57 | 82.88 | 84.90 | 85.91 |
| 14B | 73.67 | 82.83 | 83.34 | 85.35 |

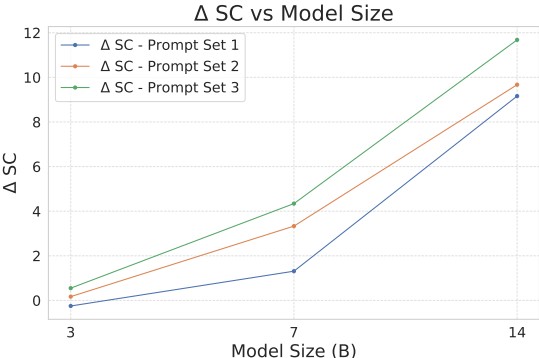

Figure 3: Trend on the change after intrinsic SC across models of different sizes.

## 6 LIMITATIONS

While the paper aims to investigate the fundamental functioning nature of intrinsic self-correction, we recognize several limitations of the presented results. Due to computational and staffing constraints, evaluations are limited to four models and six datasets. In Sections 4 and 5, the focus is mainly on a subset of models and the CommonSense QA dataset. Error bars are not provided due to resource limitations, as generating results for error analysis would require extensive computational and financial resources. While this is the common approach for the LLM community due to costs, we acknowledge that variances will exist for non-zero temperature experiments.

## 7 CONCLUSION

In this paper, we investigate the intrinsic self-correction capabilities of large language models. Supported by theoretical analysis and empirical experimentation, we present a novel perspective on the problem, positing that intrinsic self-correction is analogous to self-verification and chain-of-thought (CoT) prompting. Furthermore, we point out zero temperature and unbiased prompts are salient in facilitating intrinsic self-correction, we also underscore the significance of these two key factors. Our experimental results demonstrate that LLMs possess *universal* self-correction abilities, provided these conditions are met. Our findings will serve as a important foundation for future research and understanding into the self-correction mechanisms of large language models.

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

# A    RELATED WORKS

During LLM inference, self-correction (SC) refers to the ability to iteratively improve future responses by reflecting on the errors from previous responses. We refer readers to an excellent survey on the correction mechanism of LLMs (Pan et al., 2023). Self-correction characteristics in Large Language Models can be generally achieved by two approaches: intrinsic SC and extrinsic SC. In this section, we provide a brief survey on the current techniques used on both sides, with a focus on the intrinsic self-correction ability, which is also the focus of the main discussion of this paper.

## A.1    EXTRINSIC SC

In the current LLM community, extrinsic self-correction (SC) refers to large language models (LLMs) using external sources of knowledge to refine their own answers. This process enables the models to improve the accuracy and reliability of their responses by cross-referencing and correcting potential errors based on updated or specialized information. Such an idea could be traced back to decades ago when external information was used to guide a deep learning (DL) model to generate more accurate answers (Gou et al., 2021; Hinton et al., 2015). Since DL models were mainly used for classification tasks before generative models became prevalent, the focus was on knowledge distillation, where more powerful DL models transfer their knowledge in the training stage of smaller models. After large language models became the focus of many researchers, the idea of extrinsic self-correction started to draw attention.

Common approaches include using other LLMs as verifiers (Wang et al., 2023a; Zhang et al., 2024; Gao et al., 2023; Nathani et al., 2023; Welleck et al.), recommenders (Wang et al., 2023a; Nathani et al., 2023), or even revisers (Gao et al., 2023; Yue et al., 2024). A verifier is usually a stronger LLM (e.g. GPT4) that verifies the correctness of answers posted by a smaller and weaker LLM. For example, Zhang et al. (2024) has found that paired with a strong verifier based on GPT-4, weaker LLMs can enjoy significant accuracy improvement. A recommender suggests solutions and resources to the weaker LLMs, whereas a reviser steps in and changes the answer as needed. In many proposed works, the stronger external LLM could serve multiple roles. For example, the stronger LLM proposed by Wang et al. (2023a) both verifies the correctness of the answer and also provides suggestions for improvement.

Moreover, external self-correction often pairs up with additional features to better assist weaker models. For example, Wang et al. (2023a) finetunes the verifier with a high-quality critic dataset, and Chen et al. (2024b); Gou et al. (2024); Peng et al. (2023) propose to use retrieval augmented generation (RAG) to assist LLMs in evaluating the responses and generating higher quality criticisms and suggestions. Overall, it's the consensus in the research community that using external sources will help LLMs generate better responses during external self-correction.

## A.2    INTRINSIC SC

Albeit the success of extrinsic self-correction, researchers are more interested in the other half of the paradigm, intrinsic SC. The community's interest is partially due to the fact that extrinsic SC often requires access to an external knowledge base and even external larger LLMs. For example, although Nathani et al. (2023) shows that a multi-aspect framework, consisting of other LLMs, programming tools, and other programs, can successfully help LLMs generate better responses, the availability of those tools renders less practicality to the solution. On the other hand, if the LLMs can improve the responses solely by themselves, they will resemble the interesting self-correction ability of humans (when we make mistakes, there is a certain chance that we realize the mistake and improve our previous response without external resources).

Following the previous discussion of knowledge distillation as the predecessor of extrinsic SC, the early days of intrinsic SC could be attributed to the works of self-training, where DL models keep finetuning themselves using self-generated labels on confident predictions (Amini et al., 2024). It has been shown that if used appropriately, such an approach will also lead to more accurate DL models (Zoph et al., 2020). Intuitively, like LLMs inheriting knowledge distillation with extrinsic SC, they should also have inherited self-training with intrinsic SC. Indeed, numerous works have shown that intrinsic SC abilities exist in LLMs, with the vast majority of work using prompt engineering as the backbone (Shinn et al., 2023; Madaan et al., 2023; Li et al., 2024; Chen et al., 2024a; Liu et al.,

2024a;b; Kim et al., 2023; Dhuliawala et al., 2023; Wu et al., 2024b). Among them, Madaan et al. (2023) is usually seen as one of the first works that systematically proposes the prompt for intrinsic SC. Later works (including ours) often follow their setting, where the same LLM is used to generate feedback and improved answers to its initial responses.

However, Huang et al. (2024) challenges the existence of intrinsic SC and attributes the success of existing prompt sets to several reasons. First, Huang et al. (2024) believes that Shinn et al. (2023) and Kim et al. (2023) utilize oracle labels as the SC criteria during inference, hindering their practicality in the real world. Second, it believes that Madaan et al. (2023) uses a suboptimal prompt design that makes the initial responses unnecessarily bad. Following their arguments, they would also argue that Chen et al. (2024a), Wu et al. (2024b) and Dhuliawala et al. (2023) build suboptimal prompt sets that give favor to the refinement stages of intrinsic SC.

Besides Huang et al. (2024) and Kumar et al. (2024) who use biased prompts, in this same line of work, other negative results indicating the failure of intrinsic SC have been reported by some investigations. Jiang et al. (2024) shows that LLMs cannot discriminate among its generated content and select the best option. By sampling from multiple generation traces and asking an LLM to discriminate the best option, Jiang et al. (2024) shows that LLMs are not inherently better at discriminating (choosing the best option) among their prior outputs than they are at generating responses. Yet, Jiang et al. (2024) is under a different setting compared to intrinsic self-correction, where it uses multiple agents in the first step and uses the averaged generation results as the initial accuracy, which already brings LLM very close to its true ability. Tyen et al. (2024) argues that LLMs often struggle to identify the location of errors in CoT tracing. Yet, their results, while hinting at the lack of reasoning abilities in LLMs, show a positive net value in correcting the traces from incorrect to correct, supporting the existence of intrinsic SC ability in LLMs.

Nonetheless, as our work and some concurrent works suggest through empirical results and analyses, intrinsic SC is not a mirage. Li et al. (2024) argues on the existence through the LLM's confidence towards different questions. By designing a prompt set that lets LLM update its answers based on its confidence, that work achieves outstanding results where intrinsic SC helps improve accuracy across four different models. However, it should be noted that there is not yet a consensus on LLM's judgment over its confidence, as discussed by Singh et al. (2023) and Becker & Soatto (2024). Singh et al. (2023); Becker & Soatto (2024) indicate that LLMs show confidence in incorrect answers. From a fairness and uncertainty perspective, our work generalizes from Li et al. (2024) by taking a different path. We show that a generic fair prompt can also achieve universal intrinsic SC under zero temperature. Liu et al. (2024a) and Liu et al. (2024b) focus on a different theoretical perspective of the intrinsic SC ability. Similar to our argument, they argue that LLMs do not reach their best ability in the first round (our argument about hallucination, which they refer to as non-activated latent concepts). Through empirical results, Liu et al. (2024a) shows that intrinsic SC will cause LLM to converge to an accuracy higher than the initial performance. Their work focuses more on the convergence analysis. Instead, we conduct an analysis of the optimal conditions for LLMs performing intrinsic SC.

### A.3 Applications of Self-Correction

Besides the improvement in response accuracy, intrinsic self-correction has the potential beyond simple answer generation. Liu et al. (2023) shows that introspective reasoning can enhance the model's performance after reinforcement learning. Wang et al. (2023b) also shows that LLMs can self-generate instructions for its fine-tuning dataset. Content generated by intrinsic self-correction can be seen as reasoning traces that can be used to fine-tune the models during reinforcement learning. Such an idea is also hinted at by the growing literature on inference scaling (Kumar et al., 2024; Puerto et al., 2024). On a different line of work, due to the success of GPT4-o1, inference scaling is gaining attention in the research community (Wu et al., 2024a; Snell et al., 2024).

## B Details on Benchmarks

We use the following benchmarks:

- **CommonSenseQA** (Talmor et al., 2019) CommonSenseQA evaluates a model's ability to leverage commonsense knowledge to deduce correct answers. It comprises multiple-choice questions, each offering five possible answers, of which only one is correct.
- **GSM8K** (Cobbe et al., 2021) GSM8K is designed to assess a model's proficiency in solving arithmetic and logic problems typically encountered by elementary-level students. Each question is accompanied by a detailed step-by-step solution.
- **MMLU (Formal Logic and Conceptual Physics)** (Hendrycks et al., 2021) The MMLU benchmark comprises multiple-choice questions that test the performance of language models across a wide range of subjects. For this study, we limit our evaluation to two subjects: Formal Logic and Conceptual Physics, due to resource constraints.
- **HotpotQA** (Yang et al., 2018) HotpotQA is a question-answering dataset developed to evaluate a model's ability to engage in multi-hop reasoning. In alignment with previous studies, we utilize the version that incorporates contextual information Shinn et al. (2023).
- **BIG Bench (Sport)** (Srivastava et al., 2022) In the BIG-bench, the sports category is a collection of tasks designed to test a model's understanding and reasoning in the context of sports. The benchmark presents various scenarios, requiring the model to determine whether each scenario is feasible.
- **SVAMP** (Patel et al., 2021) SVAMP consists of simple variations of arithmetic word problems, constructed to align with a maximum difficulty of grade 4. We use the test split from Chi, which contains 300 question and answer pairs.

By default, all experiments were executed on the entire validation dataset of each benchmark. However, due to the substantial computational resources required for running the GPT-4 model across multiple temperature settings, as well as for the GSM8K benchmark, we limited the evaluation of these experiments to the first 200 questions, following the setting by Huang et al. (2024).

## C   PROOF OF PORPOSITION 2.1

Let's assume that for a comprehensive benchmark $Q$ where each question $q_i \in Q$ has $k \geq 2$ possible answers, some LLM has the true ability to answer a subset of $A \in Q$ correctly, which we denote the accuracy of its true ability as correct($A \in Q$)$=\frac{|A|}{|Q|}$. Here, the true ability is defined to reflect the knowledge of LLM's internal representation, irrespective of the generation process in practical usage. Note that the definition of true ability's accuracy is only a theoretical value and not directly measurable. Besides the true ability of LLMs (internal representation), the correctness of the answer during the generation stage is also determined by multiple factors, including decoding, prompt, temperature, etc. In addition, we assume that the LLM is able to do better than $\frac{1}{k}$ (which, from a human's perspective, is equivalent to random guessing). That is, correct($A \in Q$)$=\lambda > 1/k$.

However, since such decoding and prompting is impossible, the LLM has a hallucination possibility $h > 0$, where h% of answers are changed. Additionally, we assume that such an LLM has a Gaussian/Uniform distribution on the confidence of the answer deterministic token, which is just one token in the QA setting, for example, "Yes/No", "True/False" in a claim judgment problem or "A/B/C/D" for a multiple-choice problem. This means that hallucination randomly changes the answer (originally either correct or incorrect) with equal possibility. Such an assumption is reasonable and has been empirically shown by Becker & Soatto (2024). No matter what the previous answer was, after the hallucination, the answer is random but not the original answer. Then, we could devise the following Proposition 2.1. **Proposition 2.1.** *LLMs are generally under-performing compared to their real abilities because hallucination will cause the overall accuracy to decrease.*

*Proof of Proposition 2.1.* During the response generation, $h$ of responses are hallucinated. This means that $h * \lambda$ answers will be changed from correct to incorrect. On the other side, $\frac{h*(1-\lambda)}{k-1}$ answers will be changed from incorrect to correct.

Adding these two parts together, we get that the final accuracy after hallucination shall be $correct(A' \in Q) = \lambda - h * \lambda + \frac{h*(1-\lambda)}{k-1}$. It is not hard to see that $h * \lambda > \frac{h*(1-\lambda)}{k-1}$, as this

is the same equation of $\lambda > \frac{(1-\lambda)}{k-1} \Rightarrow \lambda * (k-1) > 1 - \lambda \Rightarrow \lambda * k > 1$, which is the assumption we make above.

Since $h * \lambda > \frac{h*(1-\lambda)}{k-1}$, we could conclude that $correct(A' \in Q) < correct(A \in Q)$. In other words, hallucination causes LLMs to perform worse. $\qquad\square$

Thus, we could model an LLM's performance on a single instance as a biased Gaussian distribution of $L \sim N(\lambda + \beta, \sigma)$, where bias $\beta$, which is related to hallucination possibility $h$, is strictly less than 0. Existing works such as self-verification and CoT are different techniques that reduce hallucination. Similarly, as argued in Section 2, SC, which is another form of self-verification and CoT, also has a similar outcome.

## D    EXPERIMENTS NOT INCLUDED IN MAIN PAPER

### D.1    NUMERICAL RESULTS FOR FIGURE 2A AND FIGURE 2B

In this section, we provide detailed numerical results for the analysis in Section 4 for the reader's reference.

#### D.1.1    NUMERICAL RESULTS FOR FIGURE 2A

Table 4 and 5 include numerical results evaluated by the GPT models on CommonSense QA dataset Talmor et al. (2019). The trend of accuracy decrement is very clear on GPT-3.5, but not so much on GPT-4 because of the response's decomposition. However, we could still see an overall downward trend for both models.

Table 4: Temperature's effect on GPT3.5 using unbiased prompt. Temperature is applied to the entire generation process.

| CommonSense QA (unbiased prompt on GPT 3.5) | | | |
|---|---|---|---|
| Temperature | Before SC | After SC | $\Delta SC$ |
| 0 | 75.92 | 76 | 0.08 |
| 0.3 | 75.02 | 75.02 | 0 |
| 0.5 | 75.51 | 74.86 | -0.65 |
| 0.8 | 74.86 | 73.96 | -0.9 |
| 1.0 | 74.45 | 72.56 | -1.89 |
| 1.2 | 74.69 | 72.4 | -2.29 |
| 1.5 | 71.5 | 69.7 | -1.8 |

Table 5: Temperature's effect on GPT-4 using unbiased prompt. Temperature is applied to the entire generation process.

| CommonSense QA (unbiased prompt on GPT 4) | | | |
|---|---|---|---|
| Temperature | Before SC | After SC | $\Delta SC$ |
| 0 | 81 | 84.5 | 3.5 |
| 0.3 | 86.5 | 86.5 | 0 |
| 0.5 | 86.5 | 87 | 0.5 |
| 0.8 | 84.5 | 84.5 | 0 |
| 1.0 | 84.5 | 84 | -0.5 |
| 1.2 | 85.5 | 85.5 | 0 |
| 1.5 | 69.5 | 69 | -0.5 |

### D.1.2 NUMERICAL RESULTS FOR FIGURE 2B

Table 6 and 7 include numerical results evaluated by the two GPT models on CommonSense QA dataset (Talmor et al., 2019) during the first ablation study. The first stage is fixed with the response generated with zero temperature. Similar to the previous study, the trend is obvious for GPT-3.5, but not for GPT-4 because of the response's decomposition.

Table 6: Temperature's effect on GPT3.5 using unbiased prompt. Temperature is applied to only the intrinsic SC process.

| CommonSense QA (unbiased prompt on GPT 3.5) | | | |
| --- | --- | --- | --- |
| Temperature | Before SC | After SC | $\Delta SC$ |
| 0 | 75.92 | 76 | 0.08 |
| 0.3 | 75.92 | 75.59 | -0.33 |
| 0.5 | 75.92 | 75.43 | -0.49 |
| 0.8 | 75.92 | 74.69 | -1.23 |
| 1.0 | 75.92 | 73.55 | -2.37 |
| 1.2 | 75.92 | 72.89 | -3.03 |
| 1.5 | 75.92 | 72.4 | -3.52 |

Table 7: Temperature's effect on GPT-4 using unbiased prompt. Temperature is applied to only the intrinsic SC process.

| CommonSense QA (unbiased prompt on GPT 4) | | | |
| --- | --- | --- | --- |
| Temperature | Before SC | After SC | $\Delta SC$ |
| 0 | 81 | 84.5 | 3.5 |
| 0.3 | 81 | 84 | 3 |
| 0.5 | 81 | 84 | 3 |
| 0.8 | 81 | 84 | 3 |
| 1.0 | 81 | 84.5 | 3.5 |
| 1.2 | 81 | 85.5 | 4.5 |
| 1.5 | 81 | 81.5 | 0.5 |

## D.2 ABLATION STUDY ON BIASED PROMPT

We provide the second ablation study of Section 4 on the effects of temperature changes on the biased prompt. As shown in Table 8 and 9, while the result seems more unstable due to the biased nature of the prompt, we see that there is still an overall trend of losing SC ability as temperature increases for GPT-3.5, whereas GPT-4 stays robust. This shows that temperature's effect is only contingent on the response's decomposition, but not on the prompt.

Table 8: Temperature's effect on GPT3.5 using biased prompt from Huang et al. (2024).

| CommonSense QA (biased prompt on GPT 3.5) | | | |
|---|---|---|---|
| Temperature | Before SC | After SC | $\Delta SC$ |
| 0 | 75.68 | 68.06 | -7.62 |
| 0.3 | 75.35 | 68.63 | -6.72 |
| 0.5 | 75.43 | 68.55 | -6.88 |
| 0.8 | 75.51 | 66.99 | -8.52 |
| 1.0 | 75.92 | 67.08 | -8.84 |
| 1.2 | 74.2 | 66.18 | -8.02 |
| 1.5 | 72.81 | 61.51 | -11.3 |

Table 9: Temperature's effect on GPT4 using biased prompt from Huang et al. (2024).

| CommonSense QA (biased prompt on GPT 4) | | | |
|---|---|---|---|
| Temperature | Before SC | After SC | $\Delta SC$ |
| 0 | 81 | 81 | 0 |
| 0.3 | 85.5 | 77.5 | -8 |
| 0.5 | 86 | 79.5 | -6.5 |
| 0.8 | 84 | 77 | -7 |
| 1.0 | 85 | 79 | -6 |
| 1.2 | 83 | 79 | -4 |
| 1.5 | 61 | 45 | -16 |

## D.3 NUMERICAL RESULTS ON THE NUMBER OF CHANGED ANSWERS

In this section, we provide detailed numerical results for the analysis on the number of changed answers for different settings. Results match the theoretical analysis that non-zero temperature and the biased prompt will cause LLMs to hallucinate and make unnecessary changes to their answers, which is detrimental to their intrinsic SC ability. As we can see, the biased prompt set (Prompt Set 1) has many more invalid changes (C2I) compared to the unbiased prompt set (Prompt Set 3), and the amount is significantly more compared to the valid changes (I2C).

### D.3.1 NOTATION REFERENCE

As mentioned above, there are five outcomes for the SC process. We provide a reference in Table 10.

Table 10: Reference sheet for the labels that will be used below.

| Label | Meaning (before SC to after SC) |
|---|---|
| C2C | correct to correct |
| C2I | correct to incorrect |
| I2C | incorrect to correct |
| I2I-C | incorrect to incorrect, but the answer changes |
| I2I-NC | incorrect to incorrect, the answer does not change |

D.3.2 CHANGE OF DISTRIBUTION OF LABELS IN RESPONSE TO TEMPERATURE CHANGES

In Table 11 to 13, we provide the number of each class for four models during the SC process as temperature changes. **It is clear that as temperature increases, the number of changed responses increases, and C2I increases much faster than I2C. This aligns with the theoretical analysis.**

Table 11: Number of changes SC makes for GPT-3.5 with unbiased prompt, with each temperature.

| CommonSense QA (unbiased prompt on GPT 3.5) | | | | |
| --- | --- | --- | --- | --- |
| Temperature | C2C | C2I | I2C | I2I-C | I2I-NC |
| 0 | 916 | 11 | 12 | 17 | 265 |
| 0.3 | 901 | 15 | 15 | 20 | 270 |
| 0.5 | 896 | 26 | 18 | 23 | 258 |
| 0.8 | 880 | 34 | 23 | 24 | 260 |
| 1.0 | 859 | 50 | 27 | 26 | 259 |
| 1.2 | 861 | 51 | 23 | 37 | 249 |
| 1.5 | 807 | 66 | 44 | 50 | 254 |

Table 12: Number of changes SC makes for GPT-4 with unbiased prompt, with each temperature.

| CommonSense QA (unbiased prompt on GPT-4) | | | | |
| --- | --- | --- | --- | --- |
| Temperature | C2C | C2I | I2C | I2I-C | I2I-NC |
| 0 | 162 | 0 | 7 | 2 | 29 |
| 0.3 | 173 | 0 | 0 | 1 | 26 |
| 0.5 | 171 | 2 | 2 | 1 | 24 |
| 0.8 | 167 | 2 | 2 | 1 | 28 |
| 1.0 | 167 | 2 | 1 | 0 | 30 |
| 1.2 | 169 | 2 | 2 | 1 | 26 |
| 1.5 | 119 | 20 | 19 | 12 | 30 |

Table 13: Number of changes SC makes for Mistral with unbiased prompt, with each temperature.

| CommonSense QA (unbiased prompt on Mistral) | | | | |
| --- | --- | --- | --- | --- |
| Temperature | C2C | C2I | I2C | I2I-C | I2I-NC |
| 0 | 803 | 3 | 11 | 12 | 392 |
| 0.3 | 833 | 4 | 5 | 13 | 366 |
| 0.5 | 837 | 3 | 3 | 9 | 369 |
| 0.8 | 836 | 6 | 4 | 20 | 355 |
| 1 | 842 | 1 | 5 | 18 | 355 |
| 1.2 | 837 | 3 | 4 | 9 | 368 |
| 1.5 | 828 | 3 | 5 | 15 | 370 |

### D.3.3 CHANGE OF DISTRIBUTION OF LABELS IN RESPONSE TO PROMPT CHANGES

In Table 14 and 15, we provide the number of each class for four models during the SC process as the prompt changes. The results validate the theoretical analysis. Biased prompt will cause more changes during the SC process. However, since the possibility of changing from a correct answer to an incorrect answer is much higher than in the other direction, the biased prompt will in fact harm the SC ability of LLM. We also get a better understanding of the models that do not suffer accuracy degradation on certain benchmarks. As we can see from I2C and C2I values in those cases, they remain very small with the biased prompt, indicating that $\gamma \simeq 0$. Such robustness might be coming from fine-tuning, or simply because the bias prompt is not "bias" enough.

Table 14: The number of occurrences of five classes of behaviors during the intrinsic SC process on the CommonSense QA dataset (Talmor et al., 2019). As we switch from a biased to an unbiased prompt, it is clear that the number of changed answers decreased significantly.

|        | GPT-3.5 | | |
| --- | --- | --- | --- |
|        | Prompt Set 1 | Prompt Set 2 | Prompt Set 3 |
| C2C    | 778 | 870 | 916 |
| C2I    | 149 | 57  | 11  |
| I2C    | 32  | 16  | 12  |
| I2I-C  | 43  | 39  | 17  |
| I2I-NC | 219 | 239 | 265 |
|        | **GPT-4** | | |
|        | Prompt Set 1 | Prompt Set 2 | Prompt Set 3 |
| C2C    | 146 | 161 | 162 |
| C2I    | 16  | 1   | 0   |
| I2C    | 16  | 8   | 7   |
| I2I-C  | 2   | 2   | 2   |
| I2I-NC | 20  | 28  | 29  |
|        | **Mistral** | | |
|        | Prompt Set 1 | Prompt Set 2 | Prompt Set 3 |
| C2C    | 775 | 805 | 803 |
| C2I    | 31  | 1   | 3   |
| I2C    | 41  | 9   | 11  |
| I2I-C  | 44  | 16  | 12  |
| I2I-NC | 330 | 390 | 392 |

Table 15: The number of occurrences of five classes of behaviors during the intrinsic SC process on the GSM8K Cobbe et al. (2021) dataset. As we switch from a biased to an unbiased prompt, it is clear that the number of changed answers decreased significantly.

| | GPT-3.5 | | |
|---|---|---|---|
| | Prompt Set 1 | Prompt Set 2 | Prompt Set 3 |
| C2C | 1015 | 1024 | 1024 |
| C2I | 9 | 0 | 0 |
| I2C | 17 | 7 | 4 |
| I2I-C | 27 | 8 | 8 |
| I2I-NC | 251 | 280 | 283 |
| | GPT-4 | | |
| | Prompt Set 1 | Prompt Set 2 | Prompt Set 3 |
| C2C | 175 | 179 | 187 |
| C2I | 14 | 10 | 2 |
| I2C | 4 | 5 | 3 |
| I2I-C | 4 | 5 | 2 |
| I2I-NC | 3 | 1 | 6 |
| | Mistral | | |
| | Prompt Set 1 | Prompt Set 2 | Prompt Set 3 |
| C2C | 570 | 597 | 606 |
| C2I | 47 | 20 | 11 |
| I2C | 51 | 29 | 26 |
| I2I-C | 224 | 112 | 107 |
| I2I-NC | 427 | 561 | 569 |

## E  PROMPT AND EXAMPLE OUTPUTS

We use the prompt formatting strategy as previous works in this area Kim et al. (2023); Shinn et al. (2023); Huang et al. (2024). We use the exact same prompt for stage 1 as Huang et al. (2024).

### E.1  PROMPT SET 1 HUANG ET AL. (2024) (BIASED) ON COMMONSENSE QA

**Q: Where do you find the most amount of leafs?**

**Answer Choices:(A) floral arrangement (B) ground (C) forrest (D) field (E) compost pile**

**Explain your reasoning. You must choose only one option from A to E. Your final answer should be a single letter from A to E, in the form (answer), at the end of your response.**

The most amount of leaves would typically be found in a (C) forest. This is because forests are densely populated with trees, which have many leaves that fall to the ground, especially during autumn when deciduous trees shed their leaves. While leaves can be found in all the other options provided, the sheer number of trees and the volume of leaves they produce in a forest would likely surpass the number of leaves found in a floral arrangement, on the ground in non-forested areas, in a field, or even in a compost pile. Final answer: (C)

**Review your previous answer and find problems with your answer.**

Upon reviewing the previous answer, one potential issue that arises is the assumption that the forest will always have the most leaves without considering the other options in certain contexts. For example, during certain times of the year or in a particularly large compost pile that has been accumulating leaves for a long time, the number of leaves in the compost pile could potentially rival or exceed the number of leaves on the forest floor. Additionally, the term ""ground"" is quite vague and could technically include the forest floor as well, which might lead to some confusion. However, despite these considerations, the reasoning provided in the original answer still holds strong for the most part. Forests are vast areas with a high density of trees, and therefore, they are likely to have a larger accumulation of leaves than the other options listed, especially when considering a typical scenario over a large area and over time. Fields generally have fewer trees and more open space, floral arrangements are small and contain a limited number of leaves, and while compost piles can have a significant number of leaves, they are usually not as vast as a forest. Final answer: (C)

**Based on the problems you found, improve your answer. You must choose only one option from A to E. Please reiterate your answer, with your final answer a single letter from A to E, in the form (answer).**

Upon further consideration, it is important to clarify that the term ""ground"" in option (B) is ambiguous and could encompass a variety of landscapes, including the forest floor. However, if we interpret ""ground"" to mean general outdoor areas that are not specifically forested, then the forest (C) still remains the most likely place to find the greatest accumulation of leaves. This is because forests are ecosystems with a high density of trees, and the natural lifecycle of these trees involves shedding leaves, particularly in deciduous forests during the fall season. While a compost pile (E) can indeed contain a significant number of leaves, it is typically confined to a smaller area compared to the expansive spread of a forest. Fields (D) are often open spaces with fewer trees and therefore fewer leaves. Floral arrangements (A) are designed and contain only a small number of leaves for aesthetic purposes. Taking into account the potential for seasonal variations and the size of compost piles, the forest still represents the most consistent and extensive source of leaves due to the sheer number of trees and the continuous process of leaf shedding and accumulation over time. Final answer: (C)

### E.2  PROMPT SET 1 HUANG ET AL. (2024) (BIASED) ON GSM8K

**Question: Can you solve the following math problem? Jill gets paid $20 per hour to teach and $30 to be a cheerleading coach. If she works 50 weeks a year, 35 hours a week as a teacher and 15 hours a week as a coach, what's her annual salary? Explain your reasoning. Your final answer should be a single numerical number, in the form \boxed{answer}, at the end of your response.**

Answer: To solve this problem, we need to calculate Jill's earnings from teaching and cheerleading coaching separately, and then add them together to find her total annual salary.

First, let's calculate her earnings from teaching: Jill gets paid $20 per hour as a teacher. She works 35 hours a week as a teacher. Her weekly earnings from teaching are: $20/hour * 35 hours/week = $700/week.

Since she works 50 weeks a year, her annual earnings from teaching are: $700/week * 50 weeks/year = $35,000/year.

Next, let's calculate her earnings from cheerleading coaching: Jill gets paid $30 per hour as a cheerleading coach. She works 15 hours a week as a coach. Her weekly earnings from coaching are: $30/hour * 15 hours/week = $450/week.

Since she also works 50 weeks a year as a coach, her annual earnings from coaching are: $450/week * 50 weeks/year = $22,500/year.

Now, we add her earnings from teaching and coaching to find her total annual salary: Total annual salary = Earnings from teaching + Earnings from coaching Total annual salary = $35,000/year + $22,500/year Total annual salary = $57,500/year.

Therefore, Jill's annual salary is \boxed{57500}.

**Review your previous answer and find problems with your answer.**

Upon reviewing the previous answer, there doesn't appear to be any problems with the calculations or the reasoning. The steps taken to calculate Jill's annual salary from both teaching and cheerleading coaching are correct, and the final answer is accurately computed by adding her earnings from both jobs.

To summarize the correct calculations:

- Annual earnings from teaching: $20/hour * 35 hours/week * 50 weeks/year = $35,000/year. - Annual earnings from coaching: $30/hour * 15 hours/week * 50 weeks/year = $22,500/year. - Total annual salary: $35,000/year (teaching) + $22,500/year (coaching) = $57,500/year.

The final answer remains \boxed{57500}.

**Based on the problems you found, improve your answer. Please reiterate your answer, with your final answer a single numerical number, in the form \boxed{answer}.**

Upon reviewing the previous answer, there is no problem with the calculations or the reasoning. The steps taken to calculate Jill's annual salary from both teaching and cheerleading coaching are correct, and the final answer is accurately computed by adding her earnings from both jobs.

To summarize the correct calculations:

- Annual earnings from teaching: $20/hour * 35 hours/week * 50 weeks/year = $35,000/year. - Annual earnings from coaching: $30/hour * 15 hours/week * 50 weeks/year = $22,500/year. - Total annual salary: $35,000/year (teaching) + $22,500/year (coaching) = $57,500/year.

The final answer remains \boxed{57500}.

E.3    PROMPT SET 2 (LESS BIASED) ON COMMONSENSE QA

**Q: If one needed the bathroom they needed a key, to get it they had to also buy something from the what?**

**Answer Choices:(A) school (B) convenience store (C) rest area (D) mall (E) theater**

**Explain your reasoning. You must choose only one option from A to E. Your final answer should be a single letter from A to E, in the form (answer), at the end of your response.**

Based on the given information, the person needing the bathroom needs a key. This suggests that the bathroom is likely locked or restricted access. To obtain the key, they have to buy something from a specific place. Out of the given options, the most likely place where one would need to buy something to obtain a key for the bathroom is a convenience store. Convenience stores often have

restrooms that are locked or require a key for access, and it is common for them to require a purchase in order to obtain the key. Therefore, the answer is (B) convenience store.

**Do you think the question and answer pair above is correct? The answer has a logical chain. If you think the answer is incorrect, please identify where the logic is wrong in the logical chain.**

The logical chain in the answer is correct. The reasoning provided is sound and aligns with common practices in many convenience stores. Therefore, the answer (B) convenience store is correct.

**Based on the responses above, answer the original question. Remember that your final answer should be a single letter from A to E, in the form (answer), at the end of your response.**

The answer to the original question is (B) convenience store.

### E.4 PROMPT SET 2 (LESS BIASED) ON GSM8K

**Question: Can you solve the following math problem? Eliza's rate per hour for the first 40 hours she works each week is \$10. She also receives an overtime pay of 1.2 times her regular hourly rate. If Eliza worked for 45 hours this week, how much are her earnings for this week? Explain your reasoning. Your final answer should be a single numerical number, in the form \boxed{answer}, at the end of your response.**

Answer: To solve this problem, we need to calculate Eliza's earnings for the first 40 hours and then calculate her overtime earnings for the additional hours worked. Eliza's regular hourly rate is \$10. For the first 40 hours, her earnings are:

40 hours * \$10/hour = \$400

For overtime, Eliza earns 1.2 times her regular hourly rate. So, her overtime rate is:

\$10/hour * 1.2 = \$12/hour

Eliza worked 45 hours in total, which means she worked 5 hours of overtime:

45 hours - 40 hours = 5 hours of overtime

Now, we calculate her earnings for the overtime hours:

5 hours * \$12/hour = \$60

Finally, we add her regular earnings to her overtime earnings to find her total earnings for the week:

\$400 (regular earnings) + \$60 (overtime earnings) = \$460

So, Eliza's total earnings for the week are \boxed{460}.

**Do you think the question and answer pair above is correct? The answer has a logical chain. If you think the answer is incorrect, please identify where the logic is wrong in the logical chain.**

The question and answer pair above is correct. The logic used to calculate Eliza's earnings for the week is sound and follows these steps:

1. Calculate regular earnings for the first 40 hours at \$10 per hour.

2. Determine the overtime rate as 1.2 times the regular rate.

3. Calculate the overtime earnings for the additional hours worked beyond 40 hours.

4. Add the regular earnings and overtime earnings to get the total earnings for the week. The calculations provided are accurate:

- Regular earnings: 40 hours * \$10/hour = \$400

- Overtime rate: \$10/hour * 1.2 = \$12/hour

- Overtime earnings: 5 hours * \$12/hour = \$60

- Total earnings: \$400 + \$60 = \$460

The final answer is correctly presented in a box as \boxed{460}.

**Based on the responses above, answer the original question. Please reiterate your answer, with your final answer a single numerical number, in the form \boxed{answer}."**

Eliza's total earnings for the week are \boxed{460}.

### E.5 PROMPT SET 3 (UNBIASED) ON COMMONSENSE QA

**Q: What do people typically do while playing guitar?**

**Answer Choices: (A) cry (B) hear sounds (C) singing (D) arthritis (E) making music**

**Explain your reasoning. You must choose only one option from A to E. Your final answer should be a single letter from A to E, in the form (answer), at the end of your response.**

People typically do (E) making music while playing guitar. Playing the guitar involves strumming or picking the strings to create melodies, chords, and rhythms, resulting in the production of music.

**Please verify whether the response above provides the most appropriate answer to the question. Provide your reasoning and justification to your verification.**

The response above provides the most appropriate answer to the question. Playing the guitar is primarily about creating music, whether it is through strumming, picking, or fingerpicking the strings. While some people may cry or experience arthritis while playing the guitar, these are not typical activities associated with playing the instrument. Hearing sounds is a passive action that can occur while playing the guitar, but it does not capture the active process of creating music. Singing can be done in conjunction with playing the guitar, but it is not the primary activity being asked about in the question. Therefore, the most appropriate answer is (E) making music.

**Based on the responses above, answer the original question. Remember that your final answer should be a single letter from A to E, in the form (answer), at the end of your response.**

People typically (E) make music while playing guitar.

### E.6 PROMPT SET 3 (UNBIASED) ON GSM8K

**Question: Can you solve the following math problem? Terry eats 2 yogurts a day. They are currently on sale at 4 yogurts for $5.00. How much does he spend on yogurt over 30 days? Explain your reasoning. Your final answer should be a single numerical number, in the form \boxed{answer}, at the end of your response.**

Answer: Since Terry eats 2 yogurts a day, he would need 2 yogurts * 30 days = «2*30=60»60 yogurts over 30 days. Since the yogurts are on sale at 4 for $5.00, Terry would need to buy 60 yogurts / 4 yogurts = «60/4=15»15 sets of 4 yogurts. Each set of 4 yogurts costs $5.00, so Terry would spend 15 sets * 5.00 =«15*5=75.00»75.00 on yogurt over 30 days. Therefore, Terry would spend $75.00 on yogurt over 30 days. Answer: \boxed{75}.

**Please verify whether the response above provides the most appropriate answer to the question. Provide your reasoning and justification to your verification.**

Yes, the response above provides the most appropriate answer to the question. The reasoning is clear and accurate. It correctly calculates the number of yogurts Terry needs over 30 days and then determines the number of sets of 4 yogurts he would need to buy. Finally, it calculates the total cost by multiplying the number of sets by the cost per set. The final answer of $75.00 is the correct amount Terry would spend on yogurt over 30 days.

**Based on the responses above, answer the original question. Please reiterate your answer, with your final answer a single numerical number, in the form \boxed{answer}.**

\boxed{75}

### E.7 SECTION 2.2 CASE 2

**Q: The hostess greeted the employees to the program, she then led them to their what? Answer Choices:(A) group people (B) welcome guests (C) occupations (D) work room (E) seat customer**

**Explain your reasoning. You must choose only one option from A to E. Your final answer should be a single letter from A to E, in the form (answer), at the end of your response.**

The hostess greeted the employees to the program, indicating that she is welcoming them and guiding them to a specific location. The most logical option would be (D) work room, as employees typically have designated areas or rooms where they carry out their tasks. Therefore, the answer is (D).

**Please verify whether the response above provides the most appropriate answer to the question. First, provide your reasoning and justification to your verification. Then, clearly state the aswer only after your reasoning and justification.**

The response above provides a reasonable justification for the answer choice (D) work room. The hostess is greeting the employees to the program, indicating that she is welcoming and guiding them to a specific location. The most logical option would be a designated work room where the employees can carry out their tasks. Therefore, the answer is (D) work room.

**Based on the responses above, answer the original question. Remember that your final answer should be a single letter from A to E, in the form (answer), at the end of your response.**

The response above provides a reasonable justification for the answer choice (D) work room. Therefore, the final answer is (D).

E.8   SECTION 2.2 CASE 3

**Q: A revolving door is convenient for two direction travel, but it also serves as a security measure at a what?**

**Answer Choices:(A) bank (B) library (C) department store (D) mall (E) new york**

**Explain your reasoning. You must choose only one option from A to E. Your final answer should be a single letter from A to E, in the form (answer), at the end of your response.**

A: bank

**Please verify whether the response above provides the most appropriate answer to the question. Provide your reasoning and justification to your verification.**

Yes, the response (A) bank is the most appropriate answer to the question. A revolving door serves as a security measure at a bank because it helps control the flow of people entering and exiting the building. The design of a revolving door allows only one person to enter or exit at a time, making it more difficult for unauthorized individuals to enter the bank quickly or without notice. Therefore, a revolving door is commonly used in banks as a security measure.

**Based on the responses above, answer the original question. Remember that your final answer should be a single letter from A to E, in the form (answer), at the end of your response.**

The most appropriate answer to the question is (A) bank.

E.9   SECTION 2.2 CASE 4

**Q: A revolving door is convenient for two direction travel, but it also serves as a security measure at a what?**

**Answer Choices:(A) bank (B) library (C) department store (D) mall (E) new york**

**Explain your reasoning. You must choose only one option from A to E. Your final answer should be a single letter from A to E, in the form (answer), at the end of your response.**

A: bank

**Please analyze the question and answer above. Provide your reasoning. Then, verify whether the response above provides the most appropriate answer to the question. Clearly state your judgement as yes or no.**

The question asks about the purpose of a revolving door as a security measure. Among the answer choices, a revolving door is commonly used as a security measure at a bank. Banks often have

revolving doors to control the flow of people entering and exiting the building, providing an added layer of security by allowing only one person to enter or exit at a time.

The response above, which selects option A: bank, provides the most appropriate answer to the question. Therefore, the judgment is yes.

**Based on the responses above, answer the original question.Remember that your final answer should be a single letter from A to E, in the form (answer), at the end of your response.**

A: bank

## F    RESPONSE DIFFERENCES BY EACH OF THE MODEL

In this section, we provide example responses of stage 2 for GPT family models. GPT-3.5 puts its final judgment in the beginning, whereas GPT-4 puts it towards the end.

### F.1    GPT-3.5

**Q: A revolving door is convenient for two direction travel, but it also serves as a security measure at a what? Answer Choices:(A) bank (B) library (C) department store (D) mall (E) new york Explain your reasoning. You must choose only one option from A to E. Your final answer should be a single letter from A to E, in the form (answer), at the end of your response.**

A: bank

A revolving door is often used as a security measure at banks. The design of a revolving door allows for controlled entry and exit, making it difficult for unauthorized individuals to enter the building. Additionally, the rotating motion of the door helps to prevent tailgating, where multiple people try to enter or exit through a single door at the same time. Therefore, the correct answer is (A) bank.

**Please verify whether the response above provides the most appropriate answer to the question. Provide your reasoning and justification to your verification.**

Yes, the response above provides the most appropriate answer to the question. A revolving door is commonly used as a security measure at banks due to its controlled entry and exit design, which helps prevent unauthorized individuals from entering the building. The rotating motion of the door also helps to prevent tailgating. Therefore, the correct answer is (A) bank.

### F.2    GPT-4

**Q: A revolving door is convenient for two direction travel, but it also serves as a security measure at a what? Answer Choices:(A) bank (B) library (C) department store (D) mall (E) new york Explain your reasoning. You must choose only one option from A to E. Your final answer should be a single letter from A to E, in the form (answer), at the end of your response.**

Revolving doors are commonly used in various types of buildings for their convenience and energy efficiency, as they help to prevent drafts, thus maintaining the interior climate. However, when considering the aspect of security, banks are the type of establishment among the options provided that are most likely to use revolving doors as a security measure. Banks often deal with higher security needs due to the nature of their business, which involves handling large amounts of money and sensitive financial transactions. While libraries, department stores, and malls may also use revolving doors for convenience and energy efficiency, they typically do not have the same level of security concerns as banks. The option ""New York"" (E) is not specific to a type of establishment but rather a location, so it does not directly answer the question about where a revolving door serves as a security measure.

Therefore, the most appropriate answer from the given options is:

(A) bank

**Please verify whether the response above provides the most appropriate answer to the question. Provide your reasoning and justification to your verification.**

The response above provides a logical and well-reasoned answer to the question. Revolving doors are indeed used in various types of buildings for their convenience and energy efficiency. However, when considering the aspect of security, banks stand out among the options provided.

Banks handle large amounts of money and are at a higher risk for theft and other security breaches. Therefore, they often implement additional security measures to protect their assets and customers. A revolving door can act as a security measure in a bank by controlling the flow of people entering and exiting the building, making it more difficult for potential robbers to quickly escape and for unauthorized individuals to enter without detection.

Libraries (B), department stores (C), and malls (D) may use revolving doors primarily for energy efficiency and convenience rather than as a primary security measure. While they do have security concerns, these are generally not as heightened as those of a bank.

Option (E) New York is not relevant to the question as it is a location and does not specify a type of establishment where a revolving door would serve as a security measure.

Given the context of the question and the options provided, the most appropriate answer is indeed:

(A) bank

This answer is justified by the higher security requirements of banks compared to the other options listed.

