# OpenReview forum: "Large Language Models have Intrinsic Self-Correction Ability"
_ICLR.cc/2025/Conference — Submitted to ICLR 2025_

### Official Review · Reviewer_shEV · 2024-10-28

**Soundness:** 1
**Presentation:** 1
**Contribution:** 1
**Rating:** 3
**Confidence:** 4

**Summary:**

- **Context: **The paper studies intrinsic self-correction which is LLMs ability to improve **their own** prior decisions (in contrast to **extrinsic** self-correction that uses external feedback of the environment). There are several prior works that cast doubt on the effectiveness of intrinsic self-correction.

 - **This presented paper:** This work argues that LLMs are indeed capable of intrinsic self-correction capabilities. The paper makes this argument through theoretical analyses and empirical experiments. The paper points out two critical factors for successful self-correction: (1)
zero temperature and (2) fair prompts.

**Strengths:**

- The problem of self-improvement is of great interest and importance for the research community, particularly given recent debates about whether LLMs possess genuine self-correction abilities.
- The issue of temperature hasn't been studied enough in the context of self-improvement. The authors' attempt to systematically analyze this parameter's influence is valuable.
- The idea to break self-improvement into multiple stages like chain-of-thought and study each component separately (as shown in Section 2.2) is interesting and potentially useful for understanding the mechanism of self-correction.

**Weaknesses:**

(1)  Theoretical Foundation Issues: The paper claims to provide theoretical proofs, but the analytical process lacks rigor. For example:
- Eq3 is just wrong, p(A|\tau) should be p(A, R | \tau). Basically, p(X, Y | Z) = p(X | Y, Z) x p(Y|Z)  (minor typo: \tau should be \tau_1). For the same reason, Eq 4 is also incorrect.
 - Sec 4.1 opens by “the randomness in decision-making diminishes as the temperature decreases”; isn’t this just a trivial statement? In Equation 8, the authors take temperature to extremes to show that answers have more likelihood to flip (which is obvious even without all these equations). But more importantly, this finding has no meaningful connection to self-improvement. Even if it does, it does connect to self-improvement, it is somewhat unrealistic since practical applications rarely use such extreme temperatures.
 - The analysis in Section 5.1 makes an unreasonable assumption that biased prompts cause random answer flipping. This assumption isn't justified, and the subsequent theoretical analysis adds little insight to our understanding.

(2) Overstatement of Results: The authors frequently jump to conclusions without sufficient evidence:
 - In line 194, they cite Table 1 as evidence of "improvement in accuracy after SC", but the difference is merely 0.08% (75.92% to 76.00%).
 - While Table 2 shows statistical significance across multiple datasets, the actual improvements are typically within 1%. This hardly constitutes meaningful improvement.
- In line 311, they make a definitive claim that "GPT-3.5 follows Order 1 of Eq. (5), whereas GPT4 follows Order 2" without providing quantitative evidence for this classification.

(3) Limited Practical Relevance of Temperature Analysis:
 - Figure 2 shows that temperature effects in the practical range (0-0.7) are minimal.
 - The significant effects only appear at temperature values like 1.4, which are rarely used in practice for self-improvement tasks.

-------------

Presentation Issues:
 - Figure 1 would benefit from including key notation (τ1, R1, D, etc.) referenced throughout the paper.
 - Critical prompt examples are relegated to the appendix when they should be in the main text for better understanding of the methodology.
- In Fig1 caption, make it clear what makes your prompt “unbiased”. One might spend a long time reading the prompt and not understanding what they should pay attention to.

> Feng et al. (2023) argues that CoT increases the effective depth of the circuit by letting the generated outputs repeatedly loop back to the input.

This phrasing is extremely misleading. The arguments of Feng et al. is that with longer generations, Transformers (for **SOME choice of weights**; not necessarily the pre-trained weights) have more expressive capacity to model more complex problems (circuits). However, this argument does not automatically mean that the existing pre-trained LLMs actually do this, which is what is implied by your phrasing.

Missing references:

There is other works that casts doubt on the abilities of LLMs to self-improve that are missing in your work:
- Self-[In]Correct: LLMs Struggle with Discriminating Self-Generated Responses, 2024
- The Google paper: LLMs cannot find reasoning errors, but can correct them given the error location, 2024
- ... (there are a few others that I might be missing)
Given that your work directly rebuts the above works, please include appropriate discission for each one (not just Huang et al.)

Here are several other relevant works that are not cited in your work:
- Crystal: Introspective reasoners reinforced with self-feedback, 2023
- Generating sequences by learning to self-correct, 2023
- The self-Instruct paper, 2023

Your work is also related to the growing literature on inference-scaling. As the prior work has shown, inference-scaling works. Naturally, one can extend these to data generation, as you do in your work.

**Questions:**

N/A

---

> ### Author Response · Authors · 2024-11-24
> **Thank you for your review! (Part 1/2)**
>
> Thank you for taking the time to review our paper and for providing valuable feedback. Your comments have helped us identify areas where further clarification and improvements were needed. We have addressed your concerns in detail below.
>
>
> > Eq3 is just wrong, p(A|\tau) should be p(A, R | \tau). Basically, p(X, Y | Z) = p(X | Y, Z) x p(Y|Z) (minor typo: \tau should be \tau_1). For the same reason, Eq 4 is also incorrect.
>
> We thank the reviewer for this careful observation, and we apologize for the incorrect equation. Indeed, it should be $p(A, R | \tau_1) $ and $p_{SC}(A′,D,R2,A,R1|τ3,τ2,τ1)$.
>
> > Sec 4.1 opens by “the randomness in decision-making diminishes as the temperature decreases”; isn’t this just a trivial statement?
>
> We agree with the reviewer that this sentence in itself may appear trivial, particularly to experts like the reviewer. However, it serves as a foundational basis for clarifying subsequent claims about the relationship between temperature increases, the likelihood of random flipping, and the degradation of self-correction ability. If we simply state, "As temperature increases, the likelihood of random flipping in the answer rises, leading to a degradation of self-correction ability," many readers may find this unclear and question the validity of the claim. The same goes for Section 4.1, which discusses the theoretical impact of temperature on decision-making.
>
>
> >In Equation 8, the authors take temperature to extremes to show that answers have more likelihood to flip (which is obvious even without all these equations).
>
> We’d like to point out that $[0,\infty)$ is the value range for T, and Equation 8 does not say that T must go to $\infty$. Rather, Equation 8 shows that on the value range of $[0,\infty)$, the derivative of variance is greater than or equal to 0, with a higher value for larger T.
>
> >But more importantly, this finding has no meaningful connection to self-improvement.
>
> We respectively disagree with this comment. The impact of T on self-improvement is indeed shown in Equation 8. As we note: “the variance of the decision increases monotonically with the model’s temperature, and the model with 0 temperature will be less likely to give the wrong decision due to less randomness,” and that “With an increasing variance, D has an increasing possibility of being flipped to the other side.” When temperature-induced flips happen, they will change more answers from correct to incorrect compared to vice versa (which is the argument of Proposition 2.1).
>
> >Even if it does, it does connect to self-improvement, it is somewhat unrealistic since practical applications rarely use such extreme temperatures.
>
> As we have suggested above, Equation 8 does not mean extreme temperatures. Rather, it means that self-correction gets worse with increasing temperature.
>
> >In line 194, they cite Table 1 as evidence of "improvement in accuracy after SC", but the difference is merely 0.08% (75.92% to 76.00%).
>
> We are very glad that you have this observation! Indeed, SC's performance looks very marginal on the Commonsense QA dataset, but this is due to the dataset's poor quality. As you may have also noticed, the improvement of chain-of-thought (CoT) on this dataset is also very marginal (75.35->75.92). CoT is a well-established and highly effective prompt engineering method, yet it exhibits only a limited effect on this particular dataset.
>
> A brief look at some of the questions and model responses in this dataset reveals its poor quality. Other than the unfiltered questions, which models typically refuse to answer due to their nature, some questions have wrong or multiple answers. Nonetheless, we present the results of this dataset to align with other works, such as Huang et al., especially since our study arrives at different conclusions. Maintaining a degree of consistency in the use of the data is essential.
>
> >While Table 2 shows statistical significance across multiple datasets, the actual improvements are typically within 1%. This hardly constitutes meaningful improvement.
>
> First and foremost, we would like to remind the reviewer that statistical significance indicates the presence of improvement. The meaningfulness of this improvement depends on the context and is not the primary focus of this work. As highlighted throughout the paper, our objective is to investigate whether intrinsic self-correction exists in LLMs. Through comprehensive analysis and experiments, we demonstrate that it does. While an improvement of 1%-2% may seem modest, it has the potential to serve as a foundation for future research aimed at amplifying this effect.

---

> > ### Author Response · Authors · 2024-11-24
> > **Thank you for your review! (Part 2/2)**
> >
> > >In line 311, they make a definitive claim that "GPT-3.5 follows Order 1 of Eq. (5), whereas GPT4 follows Order 2" without providing quantitative evidence for this classification.
> >
> > We have provided examples of each case in Appendix F. Indeed, we do not have quantitative evidence, but this is because it is hard to automatically classify the order. We admit to this limitation and will rephrase line 311 as “Based on our observations, it seems that GPT-3.5 follows Order 1 of Eq. (5), whereas GPT4 follows Order 2.”
> >
> > >Figure 2 shows that temperature effects in the practical range (0-0.7) are minimal. The significant effects only appear at temperature values like 1.4, which are rarely used in practice for self-improvement tasks.
> >
> > We agree with the reviewer that temperatures like 1.4 are rarely used in practice. The majority of use cases in practice use temperatures ranging between 0 and 1.0 with 1.0 also being the default temperature value in various frameworks. As shown in the figure, for T=1.0, the degradation of GPT3.5 is already more than 2%, which for our purposes is highly significant. Additionally, the temperatures above one are there just to show the trend.
> >
> > >Figure 1 would benefit from including key notation (τ1, R1, D, etc.) referenced throughout the paper.
> >
> > Thank you for the suggestion. Drawing the labels on the figure will make it less readable, so we explain the colors and correspond them to section 2 in the caption.
> >
> > >Critical prompt examples are relegated to the appendix when they should be in the main text for better understanding of the methodology.
> >
> > Figure 1 provides an example of the prompts used in our study. Due to space limitations, it is not feasible to include all prompts within the main text. Consistent with common practices in the majority of LLM papers, we have relegated all the prompts to the appendix.
> >
> > >In Fig1 caption, make it clear what makes your prompt “unbiased”. One might spend a long time reading the prompt and not understanding what they should pay attention to.
> >
> > Thank you for the suggestion. We have added the explanation in the caption.
> >
> > >Feng et al. (2023) argues that CoT increases the effective depth of the circuit by letting the generated outputs repeatedly loop back to the input.
> > This phrasing is extremely misleading. The arguments of Feng et al. is that with longer generations, Transformers (for SOME choice of weights; not necessarily the pre-trained weights) have more expressive capacity to model more complex problems (circuits). However, this argument does not automatically mean that the existing pre-trained LLMs actually do this, which is what is implied by your phrasing.
> >
> > We agree with your analysis of Feng et al.’s argument. However, we disagree that our phrasing is misleading. We paraphrased from Feng et al.’s paper, Section 3, last paragraph, where the original sentences were: “Actually, this can be understood via the effective depth of the Transformer circuit. By employing CoT, the effective depth is no longer L since the generated outputs are repeatedly looped back to the input. The dependency between output tokens leads to a significantly deeper circuit with depth proportional to the length of the CoT solution.” We have added your note to our discussion in the revised paper.
> >
> >
> > > Missing references:
> >
> > We thank the reviewer for those references. We have included them in the discussion of related works. Please check out the revised related work in the appendix, where we have added all of the references you mentioned and have discussed the other works that cast doubt on the abilities of LLMs to self-improve.

---

> > > ### Comment · Reviewer_shEV · 2024-11-24
> > > **Re: rebuttal**
> > >
> > > > We respectively disagree with this comment. The impact of T on self-improvement is indeed shown in Equation 8. As we note: “the variance of the decision increases monotonically with the model’s temperature, and the model with 0 temperature will be less likely to give the wrong decision due to less randomness,” and that “With an increasing variance, D has an increasing possibility of being flipped to the other side.” When temperature-induced flips happen, they will change more answers from correct to incorrect compared to vice versa (which is the argument of Proposition 2.1).
> > >
> > > It is true (and I’ve previously agreed) that higher temperatures result in more random outputs—a fairly trivial observation. However, this does not inherently link to stronger self-improvement. Increased randomness (e.g., greater variance as per Eq. 8) can simply lead to more unpredictable or noisy predictions. I’m struggling to see how these arguments offer any evidence of intrinsic self-improvement. What am I missing?

---

> > > > ### Author Response · Authors · 2024-11-24
> > > >
> > > > The review appears to have misunderstood our argument. Our main point is that increasing temperature leads to a deterioration in self-correction. In Section 4, we provide an analysis demonstrating why T=0 (lowset temperature) is the optimal setting for achieving the best self-correction.

---

> ### Comment · Reviewer_shEV · 2024-11-24
> **Re: rebuttal**
>
> > Our main point is that increasing temperature leads to a deterioration in self-correction
>
> I see. I misunderstood what you're doing here. But again, isn't this somewhat self-evident? Pretty much anything gets worse with higher temperature. Basically, you can prove a statement like "increasing temperature leads to a deterioration in X" for your favorite choice of X, just because the next token distribution becomes noisy/uninformative for higher temperature.
>
> Taking one step back, what you're proving here using temperature analysis (which takes up a good chunk of your draft; S2.2 and S4) does not seem to support with your main message,  "LLMs **have** intrinsic self-correction ability". If I am misunderstanding anything here, please let me know.

---

> > ### Author Response · Authors · 2024-11-24
> >
> > >But again, isn't this somewhat self-evident? Pretty much anything gets worse with higher temperature. Basically, you can prove a statement like "increasing temperature leads to a deterioration in X" for your favorite choice of X, just because the next token distribution becomes noisy/uninformative for higher temperature.
> >
> > It might be intuitive for an expert like you, but it is not evident for the general public studying LLMs and is debated by some works. As we said in the manuscript, “Furthermore, a very recent work from Renze & Guven (2024) has claimed to find no correlation between increasing temperature and losing accuracy on benchmarks for temperatures between 0.0 and 1.0.” Renze & Guven (2024)’s study was not on self-correction, but as we have shown in Section 4 through both theoretical analysis and experiments, temperature does affect self-correction.
> >
> >
> > >Taking one step back, what you're proving here using temperature analysis (which takes up a good chunk of your draft; S2.2 and S4) does not seem to support with your main message, "LLMs have intrinsic self-correction ability". If I am misunderstanding anything here, please let me know.
> >
> > There are two main messages from the paper. The first one is what the reviewer suggests “LLMs have intrinsic self-correction ability.” This one is easy to get, and it is evident through experiments. But there is another important message that the reviewer might miss: the optimal condition for doing self-correction studies in LLMs is under zero temperature and fair prompt. The majority of the paper analyzes why we need zero temperature and fair prompts for self-correction. If summarized in one sentence, our main message in this paper is “ LLMs have intrinsic self-correction ability, and it is best achieved under zero temperature and unbias prompts,” and that is supported by the analysis.

---

> > > ### Comment · Reviewer_shEV · 2024-11-25
> > > **Re: rebuttal**
> > >
> > > > It might be intuitive for an expert like you, but it is not evident for the general public studying LLMs and is debated by some works.
> > >
> > > I don't think this requires expert knowledge. Most NLP course discuss sampling as temperature, and cover the effect of temperature vs sampling quality.
> > >
> > >
> > > > But there is another important message that the reviewer might miss: the optimal condition for doing self-correction studies in LLMs is under zero temperature and fair prompt. The majority of the paper analyzes why we need zero temperature and fair prompts for self-correction. If summarized in one sentence, our main message in this paper is “ LLMs have intrinsic self-correction ability, and it is best achieved under zero temperature and unbias prompts,” and that is supported by the analysis.
> > >
> > > Formally what you're proving is: zero-temperature is needed for less stochasticity.
> > > I don't see how this connects to "the optimal condition for doing self-correction".

---

> > > > ### Author Response · Authors · 2024-11-26
> > > >
> > > > >I don't think this requires expert knowledge. Most NLP course discuss sampling as temperature, and cover the effect of temperature vs sampling quality.
> > > >
> > > > We respect the reviewer's expertise and their view on the perceived triviality of our analysis. While we do not wish to delve further into this argument, we would like to emphasize that some researchers hold differing perspectives from both sides.
> > > >  In addition, the effect of temperature on SC might be different from that on the normal QA setting, which is why the derivation in Section 4 needs to be included.
> > > >
> > > > >Formally what you're proving is: zero-temperature is needed for less stochasticity. I don't see how this connects to "the optimal condition for doing self-correction".
> > > >
> > > > The mainstream perception towards temperature is that higher temperature leads to more creativity and lower temperature is more deterministic, but the effect is not evident on the accuracy (within a reasonable range of T).
> > > > This is backed up by Renze & Guven (2024) in a normal QA setting with CoT.
> > > >
> > > >
> > > > To explain why a non-zero temperature does not seem to affect the quality of response in a normal QA setting between T=0 to 1.0, we assume we have a CoT response, $t_0t_1…t_n$, where $t_n$ is the final answer. Given all the previous analysis tokens, the score of $t_n$ is extremely high most of the time. Increasing temperature when the score is high has little effect on the final selection. The LLM will select the same one (the high score one) even under high temperatures.
> > > >
> > > >
> > > > This is not always the case for SC. As shown in Eq 5, there are two different composition orders in Stage 2 of SC. In the second order (rationale first then decision), it is equivalent to CoT. In fact, experiments also show that responses in this case are robust to temperature. However, there are models using the first order (decision then rationale). The temperature has a huge effect on them because the judgment is a complex action, which causes the decision to be relatively uncertain. With higher temperatures, uncertain tokens might be flipped.
> > > >
> > > >
> > > > Additionally, we’d like to provide two concrete examples using the LLM scores using the Qwen 2.5 7b model. Using CoT, the answers provided in the first stage have scores of 4.85 and 3.52. In stage 2, Qwen sometimes uses the first order and then reiterates/updates the answer at the end, where it gives judgment first, then reasoning, and finally the answer to the original questions. The scores for the judgments at the beginning of stage 2 are -1.10 and -0.82, respectively. After rationales, at the end of stage 2, Qwen answers the original question again, in which we observe scores of 7.27 and 4.23, respectively.
> > > >
> > > >
> > > > The scores signify our analysis. The decision, if made at the beginning of stage 2, is very uncertain, and is susceptible to temperature-induced flippings. Using this analysis, our paper presents that the optimal temperature for SC should be 0, as we cannot fully control the second stage’s order.

---

### Official Review · Reviewer_6ZBp · 2024-11-03

**Soundness:** 3
**Presentation:** 3
**Contribution:** 1
**Rating:** 5
**Confidence:** 4

**Summary:**

The paper investigates the capacity of LLMs to perform self-correction without relying on external sources of information, namely “intrinsic self-correction.” The authors propose that this self-correction process is valuable for enhancing LLM performance and has similar principles as chain-of-thought and self-verification. The study identifies two key factors that influence self-correction: using an unbiased prompt to avoid influencing the model’s initial answer and setting a zero temperature to reduce randomness. Through theoretical analysis and empirical experiments, the paper demonstrates that intrinsic self-correction is achievable across multiple LLMs, providing a foundation for further research in leveraging this capability.

**Strengths:**

1. **Insightful Analysis of Self-Correction Mechanisms**: The paper offers a novel perspective by comparing intrinsic self-correction to chain-of-thought and self-verification techniques. This theoretical framing provides a solid foundation for understanding the mechanisms that enable LLMs to self-correct.
2. **Practical Recommendations for Enhanced Model Performance**: By identifying unbiased prompts and zero temperature as key factors for effective self-correction, the authors present valuable insights that are directly applicable to real-world LLM deployment.

**Weaknesses:**

1. **Limited Performance Improvement Even in Ideal Conditions**: The performance gain from self-correction is relatively small, even under the ideal settings proposed (unbiased prompts and zero temperature). As is shown in Table 1 & 2, under many circumstances, the performance gain is less than 2%. This marginal improvement raises questions about the practical impact of the self-correction process and the techniques provided in the paper.
2. **Lack of Significant Theory and Insight**: The paper creatively related the performance of self-correction to chain-of-thought prompting and temperature, but it fails to introduce solid theoretical backbones or experimental insights.
3. **Limited Model Sizes and Architectures**: The paper only conducted experiments on two API-based models and two open-source models, and it fails to conduct ablation on model sizes (using the same series of models with different sizes) and other model architectures.   This limitation restricts the generalizability of the findings across a broader range of model types.

**Questions:**

1. What are the novel contributions of this paper over previous works on intrinsic self-correction? Given that the paper investigates the topic related to prior work on self-correction (e.g., [Pan et al., 2023](https://arxiv.org/abs/2308.03188)), it would be helpful to clarify how this research extends beyond or differs from earlier findings.
2. Why is some certain experimental results not obviously show the advantage of self-correction? Is it from the method itself, or there are other failure modes?

---

> ### Author Response · Authors · 2024-11-24
> **Thank you for your review! (Part 1/2)**
>
> Thank you for taking the time to review our paper and for providing valuable feedback. Your comments have helped us identify areas where further clarification and improvements were needed. It seems the listed Weakness is explained more in detail in Questions, so we will address the questions raised by you. If you have further confusion, please let us know in subsequent discussions. We have addressed your concerns in detail below.
>
> > What are the novel contributions of this paper over previous works on intrinsic self-correction? Given that the paper investigates the topic related to prior work on self-correction (e.g., Pan et al., 2023), it would be helpful to clarify how this research extends beyond or differs from earlier findings.
>
> We provide a review of related works in Appendix A. The survey by Pan et al. is a very good summary of the big field of LLM answer correction, and under their characterization, our work falls into post-hoc correction-> self-correction (section 5.1 of their paper). We have included the reference of this survey paper in our revised related works to direct future readers to this excellent piece of work.
>
> Regarding the novelty of our work, as we mentioned in the introduction, there have been recent debates on whether the intrinsic self-correction ability of LLM exists. Huang et. al’s work basically claimed that all the previous intrinsic self-correction papers were not under fair experiments. According to our observation, this debate on the existence of intrinsic self-correction in LLMs is primarily because the field of intrinsic self-correction lacks theoretical analysis, and people can always debate in experimental settings. Pan et al. also noted this in their future work section, urging for more theoretical foundations of this topic (their paper Section 7.1). As we highlighted in our list of contributions (first three points), our novelty is to provide theoretical analyses on this topic and give guidelines on the experiment setting. In addition to theoretical analyses, we also provide empirical experiments to show that what we say is true.
>
> Our work argues this problem through the prompt-and-answer decomposition, as well as the connection with CoT and self-verification. There are several concurrent works that came out in the same period as our work. They tackle this problem from different angles, and we have discussed them in the related works section as well. “.Li et al.(2024) argue on the existence through the LLM’s confidence towards different questions,” and “Liu et al.(2024a) and Liu et al.(2024b) focus on a different theoretical perspective of the intrinsic SC ability….Their work focuses more on the convergence analysis.”
>
> We additionally provide an analysis of the effects of temperature and prompt design on intrinsic self-correction. To the best of our knowledge, no previous works systematically studied temperature’s effect on self-correction. While there have been a lot of different prompt designs from previous works (and those are the root of the debate on the existence of intrinsic self-correction), we analyze the effects of semantic meaning and structure of prompts on intrinsic self-correction, which was not done previously.
>
> > Why is some certain experimental results not obviously show the advantage of self-correction? Is it from the method itself, or there are other failure modes?
>
> This is expected. As we try to argue in the paper, self-correction is just an implicit self-verification. When the model is already good at answering, it can only occasionally correct its mistakes in a dataset, and hence, there are no expectations of having big differences before and after intrinsic self-correction. We have said in Lines 454-456 that “Importantly, our results align with the self-verification results obtained by  Weng et al. (2023)  where they also obtained a slight improvement using a different model.”
>
> So to directly answer the question. This is from two parts, the method itself and today’s models. However, this does not mean the intrinsic self-correction method is not useful. The vanilla intrinsic self-correction does not have a big difference, but as long as it exists, there have been and will be more future works to amplify its effects.
>
> > Limited Model Sizes and Architectures:
>
> We are happy to provide more results supporting our claim. We additionally conduct experiments on three benchmarks related to math and logic on Qwen 3b,7b, and 14b using the same settings as other experiments. We provide a snapshot of the new results below:

---

> > ### Author Response · Authors · 2024-11-24
> > **Thank you for your review! (Part 2/2)**
> >
> > #### Table: Evaluation of Math and Logic Benchmarks on the Qwen 2.5 Family We evaluate three benchmarks related to math and logic on the Qwen 2.5 family to study the model size's effect on intrinsic self-correction.
> > ## GSM8K
> > | Model | Before SC | Prompt Set 1 | Prompt Set 2 | Prompt Set 3 |
> > |-------|-----------|--------------|--------------|--------------|
> > | 3B | 85.22 | 85.22 | 85.32 | 85.52 |
> > | 7B | 83.55 | 87.19 | 88.55 | 88.93 |
> > | 14B | 74.53 | 86.20 | 86.13 | 88.55 |
> >
> > ## MMLU (Formal Logic and Conceptual Physics)
> > | Model | Before SC | Prompt Set 1 | Prompt Set 2 | Prompt Set 3 |
> > |-------|-----------|--------------|--------------|--------------|
> > | 3B | 57.34 | 57.06 | 58.17 | 59.00 |
> > | 7B | 69.25 | 64.27 | 67.31 | 70.36 |
> > | 14B | 66.94 | 71.19 | 68.98 | 71.75 |
> >
> > ## SVAMP
> > | Model | Before SC | Prompt Set 1 | Prompt Set 2 | Prompt Set 3 |
> > |-------|-----------|--------------|--------------|--------------|
> > | 3B | 88.33 | 87.00 | 88.00 | 88.67 |
> > | 7B | 87.67 | 86.33 | 90.00 | 91.33 |
> > | 14B | 78.00 | 82.00 | 88.33 | 87.67 |
> >
> > ## Average
> > | Model | Before SC | Prompt Set 1 | Prompt Set 2 | Prompt Set 3 |
> > |-------|-----------|--------------|--------------|--------------|
> > | 3B | 80.61 | 80.36 | 80.78 | 81.16 |
> > | 7B | 81.57 | 82.88 | 84.90 | 85.91 |
> > | 14B | 73.67 | 82.83 | 83.34 | 85.35 |

---

> > > ### Comment · Reviewer_6ZBp · 2024-11-24
> > > **Thanks for your reply**
> > >
> > > Thank you to the authors for their detailed response! While the rebuttal provides some additional information, I believe it does not sufficiently change my perspective regarding the novelty and contribution of the work. Therefore, I will maintain my original score.

---

> > > > ### Author Response · Authors · 2024-11-24
> > > >
> > > > We respectfully request that the reviewer provides a more detailed explanation of their perspective regarding the (lack of) novelty and contributions of our work.
> > > >
> > > > The question regarding novelty is quite broad, especially considering that we have addressed this point in our prior responses. To further clarify, our work specifically provides **theoretical analyses** that contribute to a deeper understanding of intrinsic self-correction, as well as **practical guidelines for experimental settings**.
> > > >
> > > > Regarding the survey paper referenced (Pan et al., 2023), we emphasize that it explicitly highlights a gap in existing research, stating:
> > > >
> > > > >Although LLMs have exhibited a remarkable capability for self-analysis and self-improvement, there remains a lack of theoretical justifications to uncover the mystery of such ability. Therefore, we argue that the study of underlying theoretical principles can offer a more transparent understanding of self-correction.
> > > >
> > > > Our work directly addresses this gap by contributing the theoretical insights that Pan et al. call for, going beyond descriptive comparisons by providing a foundational understanding of self-correction mechanisms.
> > > > We kindly ask the reviewer to reconsider their evaluation and provide more specific feedback on what additional aspects of novelty or theoretical contributions they expect. This would enable us to further strengthen the paper in a targeted manner.

---

### Official Review · Reviewer_3P3T · 2024-11-04

**Soundness:** 1
**Presentation:** 1
**Contribution:** 1
**Rating:** 3
**Confidence:** 5

**Summary:**

This paper investigates the sefl-correction ability of LLMs. In particular, they investigate two factors, i.e. temperature and the bias of prompt. The authors conduct experiments on 6 benchmarks using 4 LLMs. However, the paper is very difficult to read, with less rigour proof. The selection of models is not wide, resulting in less convincing empirical results. Also, I do not see a systematic and coherent motivation to investigate temperature and the bias of the prompt. It is somehow more like assembling these two factors in one paper.

**After the rebuttal**: the definitions of concepts in this paper are very unclear, with many left unexplained. Most proofs are not valid and rigorous. Many sections are difficult to understand. While I defer to some reviewer's appreciation, I do not believe this paper merits an ICLR paper standard.

**Strengths:**

May be interesting to future research.

**Weaknesses:**

1. The paper is very difficult to read. In particular, the equation part.  Proof 2.1 is very confusing and many explanations are not given, and I cannot understand it with 2 or 3 times reading. For example, in Line 869, "which we denote as correct(A ∈ Q) = λ > k1". What is λ? The portion of correctly answered questions/all questions? And I do not think the probability of that hallucination randomly changing the answer is equal: tokens are of different importance to a sentence (answer), thus, some hallucinated tokens may not change the final answer (this is also related to how the authors define hallucination, but I do not see any justification here). Also, what is h? The portion of hallucinated responses? I also do not understand what the authors mean by h∗(1−λ)/(k−1) answers will be changed from incorrect to correct. Also, why h∗λ> h∗(1−λ)/(k-1)? Line 283, "(a1/T −(1−α)1/T)" should be "α" not "a".
2. The selection of models is limited. More open sourced model should be included for a more comprehensive evaluation.

**Questions:**

Please see above.

---

> ### Author Response · Authors · 2024-11-24
> **Thank you for your review! (Part 1/2)**
>
> Thank you for taking the time to review our paper and for providing valuable feedback. Your comments have helped us identify areas where further clarification and improvements were needed. We have addressed your concerns in detail below.
>
> >The paper is very difficult to read. In particular, the equation part. Proof 2.1 is very confusing and many explanations are not given
>
> We apologize for any confusion caused by our theoretical analysis. We will try to help you understand them by answering your confusion.
>
> >in Line 869, "which we denote as correct(A ∈ Q) = λ > k1". What is λ? The portion of correctly answered questions/all questions?
>
> Yes, your understanding is correct. We updated the definition of  λ in Line 868 to make it clear as “...some LLM has the true ability to answer (and not purely guess) a subset A∈Q correctly, which we denote its accuracy as correct(A ∈ Q)=$\frac{|A|}{|Q|}=λ>\frac{1}{k}$.”
>
> >And I do not think the probability of that hallucination randomly changing the answer is equal: tokens are of different importance to a sentence (answer), thus, some hallucinated tokens may not change the final answer (this is also related to how the authors define hallucination, but I do not see any justification here).
>
> Your claim on the difference in token importance in the sentence is correct, but in the proof, we are analyzing the answer deterministic token, which is just one token in the QA setting, for example, “Yes/No”, “True/False” in a claim judgment problem or “A/B/C/D” for a multiple-choice problem. We have made this clear in the revised paper.
>
> >Also, what is h? The portion of hallucinated responses?
>
> Yes, your understanding is correct. We define h as the percentage of LLM’s key answer deterministic token that is changed. We add a sub-clause after the definition of h as “where h% of answers are changed.”
>
> >I also do not understand what the authors mean by h∗(1−λ)/(k−1) answers will be changed from incorrect to correct.
>
> First, in Lines 872-875, we provide our assumption that the LLM’s responses’ selected logits (confidence) are in a Uniform/Gaussian distribution, following the analysis made by Becker & Roberto (2024). Then, since we have h% of answers changed, and the accuracy of LLM is λ, the expected value of incorrect answers that are changed is h∗(1−λ). Since there are k answer choices and incorrect answers must originally be chosen from an incorrect answer choice, for each question, the answer change has k-1 choices, of which one is the correct answer. Thus, in total, the expected number of answers that are changed from incorrect to correct is (the expected value of incorrect answers that are changed)*(the chance that it gets changed to correct) = h∗(1−λ)/(k-1).
>
> > Also, why h∗λ> h∗(1−λ)/(k-1)?
>
> First, we will reiterate what is written in Line 888-890:
> It is not hard to see that $h*\lambda > \frac{h*(1-\lambda)}{k-1}$, as this is the same equation of $\lambda>\frac{(1-\lambda)}{k-1} \Rightarrow \lambda*(k-1)>1-\lambda \Rightarrow \lambda*k>1$ which is the assumption we make above.
>
> Now, we show a detailed step-by-step analysis of how we get the result:
>
> $h*\lambda > \frac{h*(1-\lambda)}{k-1}$
>
> $\lambda > \frac{(1-\lambda)}{k-1}$ $\quad\quad$ divide each side by h
>
> $\lambda ∗ (k − 1) > 1 − \lambda $ $\quad\quad$multiple each side by k-1, since k>=2, the sign is not flipped
>
> $\lambda ∗ k > 1$	$\quad\quad$	expand the equation on the left and each side + λ
>
> In Line 869, we stated that λ > 1/k, which is the same as the derived result above.
>
> > Line 283, "(a1/T −(1−α)1/T)" should be "α" not "a".
>
> Thank you for pointing this out. We have corrected it in the revision.
>
> > The selection of models is limited. More open sourced model should be included for a more comprehensive evaluation.
>
> Before we provide more results, we want to point out that Huang et al’s paper was accepted by ICLR with 3 models and 3 benchmarks. We are not saying that’s the “golden” standard (as there was no standard), but we just also hope the reviewers can be more generous in the critiques on the number of models as they can never be enough..
>
> We have already provided 4 models and 6 benchmarks. Nonetheless, we are happy to provide more results supporting our claim. We additionally conduct experiments on three benchmarks related to math and logic on Qwen 3b,7b, and 14b using the same settings as other experiments. We provide a snapshot of the new results below:

---

> > ### Author Response · Authors · 2024-11-24
> > **Thank you for your review! (Part 2/2)**
> >
> > #### Table: Evaluation of Math and Logic Benchmarks on the Qwen 2.5 Family We evaluate three benchmarks related to math and logic on the Qwen 2.5 family to study the model size's effect on intrinsic self-correction.
> > ## GSM8K
> > | Model | Before SC | Prompt Set 1 | Prompt Set 2 | Prompt Set 3 |
> > |-------|-----------|--------------|--------------|--------------|
> > | 3B | 85.22 | 85.22 | 85.32 | 85.52 |
> > | 7B | 83.55 | 87.19 | 88.55 | 88.93 |
> > | 14B | 74.53 | 86.20 | 86.13 | 88.55 |
> >
> > ## MMLU (Formal Logic and Conceptual Physics)
> > | Model | Before SC | Prompt Set 1 | Prompt Set 2 | Prompt Set 3 |
> > |-------|-----------|--------------|--------------|--------------|
> > | 3B | 57.34 | 57.06 | 58.17 | 59.00 |
> > | 7B | 69.25 | 64.27 | 67.31 | 70.36 |
> > | 14B | 66.94 | 71.19 | 68.98 | 71.75 |
> >
> > ## SVAMP
> > | Model | Before SC | Prompt Set 1 | Prompt Set 2 | Prompt Set 3 |
> > |-------|-----------|--------------|--------------|--------------|
> > | 3B | 88.33 | 87.00 | 88.00 | 88.67 |
> > | 7B | 87.67 | 86.33 | 90.00 | 91.33 |
> > | 14B | 78.00 | 82.00 | 88.33 | 87.67 |
> >
> > ## Average
> > | Model | Before SC | Prompt Set 1 | Prompt Set 2 | Prompt Set 3 |
> > |-------|-----------|--------------|--------------|--------------|
> > | 3B | 80.61 | 80.36 | 80.78 | 81.16 |
> > | 7B | 81.57 | 82.88 | 84.90 | 85.91 |
> > | 14B | 73.67 | 82.83 | 83.34 | 85.35 |

---

> ### Comment · Reviewer_3P3T · 2024-11-24
>
> To make $h*\lambda > \frac{h*(1-\lambda)}{k-1}$ hold true, $\lambda > 1/k$.
> But I do not see why  λ > 1/k is always true.

---

> > ### Author Response · Authors · 2024-11-24
> >
> > $\frac{1}{k}$ is equivalent to the accuracy of random guessing. If LLM is properly trained and possesses some knowledge of Q, then it should do better than random guessing. If $\lambda \leq \frac{1}{k}$, it simply means the LLM does not have any idea of what it is answering, and this is not under the scope of the discussion in this paper.

---

> > > ### Comment · Reviewer_3P3T · 2024-11-24
> > >
> > > While  $\lambda > 1/k$ indicates no understanding of the question, there is no theoretical or empirical justification provided to support this assumption universally.
> > > Furthermore, $\lambda > 1/k$ is not well discussed in the paper, weak LLMs may fail on challenging datasets. Since the proof and the conclusion of this paper heavily depend on  $\lambda > 1/k$, the authors should have made it more clear.
> > > Also, it indicates its generality and applicability are not universal.

---

> > > > ### Author Response · Authors · 2024-11-24
> > > >
> > > > We assume it’s a typo for the first two λ>1/k in the reviewer’s comment. It should be λ<1/k.
> > > >
> > > > We want to point out that the emphasis of this paper is on self-correction. Self-correction literally means correcting oneself from mistakes, and in the case of intrinsic, it means doing so without external help. An example of this is when someone says something wrong and then realizes the mistake and corrects his sayings. In order to realize and correct such mistakes, humans need to have the knowledge of the correct answer! If someone does not even know the correct answer in his knowledge, how can he self-correct and get the correct answer?
> > > >
> > > > The same applies to the concept of self-correction in LLM.
> > > >  In the case of LLM, if λ<1/k and the model cannot answer challenging datasets, then it cannot improve through self-correction because the premise of knowledge is missing.    λ>1/k is a requirement of self-correction by definition.

---

> > > > > ### Comment · Reviewer_3P3T · 2024-11-25
> > > > >
> > > > > First, I thank the authors for the typos.
> > > > >
> > > > > Second, I would like to point out that it is the author's responsibility to make their paper sound and accurate.
> > > > > In particular, they should be rigorous since they claim they are "proving" something.
> > > > >
> > > > > Also, the example that the authors just provide is instance-wise, what they claim in the paper is dataset-wise. There is a difference.
> > > > > Assuming there is a dataset of m common sense QA questions (2-choice, \( 1/k = 0.5 \)), and assume the LLM has the true ability to answer n questions ( $\lambda = n/m$%, i.e., $correct(A∈Q)$)  (Line949) and no knowledge of the remaining $(m-n)$ questions (assuming LLM randomly guesses on them, $(m-n)/2$ are correct).
> > > > > After hallucination (assuming $h$, $0<h<1$), $n-nh$ initially correct answers (the true ability ones) remain correct, and $(m-n)/2$ out of the $m-n$ instances (no knowledge ones) are correct (since LLMs have no knowledge, still randomly guess).
> > > > > The final accuracy is $(n+m-2hn)/2m$ (according to the paper, this is $correct(A′ ∈Q)$, Line966-967).
> > > > > Apparently, the relation between $correct(A′ ∈Q)$ and  $correct(A∈Q)$ depends on $n$, $m$ and $h$.
> > > > >
> > > > > Kindly let me know if I miss something.

---

> > > > > > ### Author Response · Authors · 2024-11-25
> > > > > >
> > > > > > Your derivation is accurate regarding post-hallucination accuracy, but there is an error in your pre-hallucination accuracy calculation. If the LLM can answer n out of m questions correctly and randomly guess half of the remaining m-n questions, then its accuracy before hallucination is $n/m+(m-n)/2m = (n+m)/2m$.
> > > > > >
> > > > > > Then, compared to this pre-hallucination accuracy, since h is positive, we hope that it is clear $\frac{n+m}{2m} > \frac{n+m−2hn}{2m}$, regardless of m and n.

---

> ### Comment · Reviewer_3P3T · 2024-11-25
>
> According to the definition in Line 948-949: ```some LLM has the true ability to answer (and not purely guess) a subset of A ∈ Q correctly```, A does not contain ```purely guess``` answers so |A| is simply $n$. And according to their calculation, $correct(A \in Q) = \frac{|A|}{|Q|} = n/m$.
> With the step-by-step calculation, I hope that it is clear the authors realise their errors in their presentation and be more careful with the claims and proofs in their submission.

---

> > ### Author Response · Authors · 2024-11-25
> >
> > We apologize for this confusion in writing. What we meant with "(not purely guess)" is that the LLM is not purely guessing the entire Q, and this phrase is used to define $\lambda>\frac{1}{k}$.
> >
> > A does contain guess answers.
> >
> > We propose to revise the definition as follows:
> >
> > Let's assume that for a comprehensive benchmark $Q$ where each question $q_i\in Q$ has $k>=2$ possible answers, some LLM has the true ability to answer (regardless of guessing or solving) a subset of $A\in Q$ correctly, which we denote its accuracy as $correct(A\in Q) = \frac{|A|}{|Q|}$. In addition, we assume that the LLM can solve at least a subset of questions and is not purely guessing the whole dataset, which means that $correct(A\in Q) = \lambda > \frac{1}{k}$.
> >
> > Please let us know if this revision makes it clearer, and if so, we will update the draft.

---

> > > ### Comment · Reviewer_3P3T · 2024-11-25
> > >
> > > First, can the authors be more clear with ```has the true ability to answer``` - can randomly guessing be regarded as ```true ability```?
> > >
> > > Second, such "proof" mainly focuses on multi-choice QAs. Is it still applicable for tasks such as GMS8k?

---

> > > > ### Author Response · Authors · 2024-11-26
> > > >
> > > > > First, can the authors be more clear with has the true ability to answer - can randomly guessing be regarded as true ability?
> > > >
> > > > We greatly appreciate your continuous suggestions, which are extremely valuable to improving our manuscript. Please accept our sincere gratitude for your comments.
> > > >
> > > > Let’s take our discussion one step back. We apologize for the usage of “guess” in our definition (though our intent is to say that $\lambda >1/k$, where 1/k is essentially guessing from a human’s perspective). However, technically speaking, LLMs do not have the notation of guessing or solving. It is a probabilistic model that autoregressively predicts the next tokens until it reaches the answer token (which is what this proposition analyzes). Ultimately,  LLM can only answer a question correctly or incorrectly. In addition to the knowledge of LLM, which is encoded in its internal representation/weights, the correctness of the answer is also determined by multiple factors, including decoding, prompt, temperature, etc.
> > > >
> > > > That being said, the true ability of LLM is defined as its potential to correctly answer a subset of the questions in dataset Q. That is, the true ability is based on LLM’s internal representation and not directly expressible in terms of the LLM’s generation quality. In Proposition 2.1, we argue that during the evaluation of LLM on Q, there will be hallucinations where the final answer is different from what the internal representation actually wants to express.
> > > >
> > > > If the reviewer finds this explanation acceptable and clear, we will update Proposition 2.1 accordingly. We will update it to:
> > > >
> > > > Let's assume that for a comprehensive benchmark $Q$ where each question $q_i∈Q$ has $k \geq 2$ possible answers, some LLM has the true ability to answer a subset of A∈Q correctly, which we denote the accuracy of its true ability as correct(A∈Q)=$\frac{|A|}{|Q|}$. Here, the true ability is defined to reflect the knowledge of LLM’s internal representation, irrespective of the generation process in practical usage. Note that the definition of true ability’s accuracy is only a theoretical value and not directly measurable. Besides the true ability of LLMs (internal representation), the correctness of the answer during the generation stage is also determined by multiple factors, including decoding, prompt, temperature, etc. In addition, we assume that the LLM is able to do better than 1/k (which, from a human’s perspective, is equivalent to random guessing). That is, correct(A∈Q)=$\lambda>1/k$.
> > > >
> > > >
> > > > >Second, such "proof" mainly focuses on multi-choice QAs. Is it still applicable for tasks such as GMS8k?
> > > >
> > > > Yes, most evaluation benchmarks, as long as there are finite true answers for each question (for math, there is only one), can be seen as multiple-choice questions. It’s just that instead of fixed choices, you can think that the number of choices (and the content of choices) is dynamic, depending on the LLM’s logits.

---

### Official Review · Reviewer_mqrf · 2024-11-09

**Soundness:** 4
**Presentation:** 3
**Contribution:** 4
**Rating:** 10
**Confidence:** 3

**Summary:**

The paper claims to demonstrate the intrinsic self-correction capabilities of LLMs. Motivated from examples where bias in the prompts can interfere with the correction (in particular, by implying there’s a problem with the initial answer), they ask *if they can correct a wrong answer, why can’t an LM get it right the first time?*

They claim that they can’t due to inherent hallucinations but can perform intrinsic SC. They provide some theoretical analysis to demonstrate this, alongside empirical investigations into the impact of temperature, prompt structure, etc.

According to their theoretical analysis, using CoT during the initial answer generation produces a rationale alongside it, which can be attended over in the self-correction step to produce a candidate decision before reasoning for the decision has to be written. However, without the initial chain, generating this decision is effectively relying on another zero-shot correction in line with how the initial answer was (possibly erroneously) generated. In other words, systems where answers are generated by CoT are more robust to self-correction error introduction than ones that don’t use it, and proper self-correction on non-CoT answers essentially just brings performance back up to CoT levels.

This analysis leads to predictions such as hallucination rate increasing with higher temperature leading to worse performance of self-correction. This is confirmed. They also make some claims that feel more dubious to me, like “LLMs are naturally underperforming their intrinsic ability” which feels a little underdetermined to me.

**Edit**: I am happy with this paper overall, and will defer somewhat to the other reviewers on gripes about the theoretical arguements. That being said, I appreciate the authors' efforts to expand the scope of their experiments in the rebuttal.

**Strengths:**

Solid theoretical argument that runs throughout the paper. Though this isn’t my core area it was quite easy to understand and most claims were plausible.

The experiments are well connected to the theoretical analysis. Neither part feels like an afterthought, which is a bit rare for LLM reasoning papers :)

The discovery of best practices for prompt sets can both be directly applied in engineering work and used to inspire future analysis.

**Weaknesses:**

Few, mainly the more philosophical claims I complained about in my summary.

**Questions:**

I wasn’t the most well-positioned to assess the novelty of this work. I took a look through Pan et al. 2024’s survey “Automatically Correcting Large Language Models” paper to help check this, might be useful to reference for other readers?

---

> ### Author Response · Authors · 2024-11-24
> **Thank you for your review and support!**
>
> We are extremely grateful for the reviewer’s unambivalent rating of 10 for our work. We hope there will be more readers like you who can appreciate the potential significance of our work in providing “best practices for prompt sets” that “can both be directly applied in engineering work and used to inspire future analysis.” Thank you.
>
> Below, we will address your concerns.
>
> >They also make some claims that feel more dubious to me, like “LLMs are naturally underperforming their intrinsic ability” which feels a little underdetermined to me.
>
> We agree with you that the statement (“LLMs are naturally underperforming their intrinsic ability”) was not very precise, and indeed it was almost impossible to quantify LLM’s true intrinsic ability. What we meant was that the LLM’s true intrinsic ability (whatever it was) can be considered as the theoretical upper bound of its knowledge. We argue in our paper that, such a true intrinsic ability cannot be reflected through its answers because of potential hallucinations during decoding.
>
> Our results are in line with your intuition as well. We argue that, through intrinsic self-correction, we bring LLM’s output closer to its theoretical upper bound. This is essentially what other prompt engineering methods try to do. That’s the beauty of our paper’s insights and our unique contribution. Thank you for seeing this so clearly.
>
> >I wasn’t the most well-positioned to assess the novelty of this work. I took a look through Pan et al. 2024’s survey “Automatically Correcting Large Language Models” paper to help check this, might be useful to reference for other readers?
>
> Thank you. We have included this very much relevant review paper in our revision.

---

> > ### Comment · Reviewer_mqrf · 2024-12-03
> > **Rebuttal acknowledged!**
> >
> > Thank you for answering my concern there! Please consider slightly weakening statements like the intrinsic ability one we covered above in the CR if it gets accepted.

---

### Meta-Review · Area_Chair_e1DH · 2025-01-03

**Metareview:**

This paper studies intrinsic self-correction in large language models (LLMs) – their ability to revise previous answers without external input. looking at how temperature and the bias of prompts affect their performance. The paper’s core claim is that with carefully chosen “fair” prompts and zero temperature at sampling, LLMs can correct some of their initial mistakes. The paper includes theoretical analysis and empirical experiments conducted across multiple benchmarks and LLMs. The main conclusion is that intrinsic self-correction can provide small but consistent performance gains.

After evaluating the reviews and the authors’ subsequent clarifications, I am inclined to side with Reviewers 3P3T and shEV, who both raise substantial issues regarding clarity, rigor, and limited practical significance. I agree with Reviewer shEV that temperature analysis does not provided a meaningful contribution and also that the reported gains in typical temperature range are minimal.

While the authors make a good-faith effort to address questions, multiple reviewers remain unconvinced that the paper demonstrates the rigor and impact required for acceptance.

**Additional Comments On Reviewer Discussion:**

Two of the reviewers (shEV and 3P3T) provided thorough and informed comments and engaged with the authors for clarification and discussion. Reviewer shEV pointed out that the temperature analysis is fairly trivial and only weakly supports the main claim, as well as limited practical relevance of the results, given small gains in the usual temperature range. There is also some missing discussion of related work, as pointed out by shEV.  Reviewer 3P3T criticized the clarity and rigor of the theoretical proofs, pointing out confusion in definitions and assumptions and limited empirical scope. The authors made commendable efforts to address the comments and expand the scope of their experiments in the rebuttal, but I am afraid the main concerns did not go away.

---

### Decision · Program_Chairs · 2025-01-22

Reject